# Decomposing oceanic temperature and salinity change using ocean carbon change

Charles E. Turner[1], Peter J. Brown[2], Kevin I. C. Oliver[1], and Elaine L. McDonagh[2,3]

[1]University of Southampton, European Way, Southampton, SO14 3ZH
[2]National Oceanography Centre, European Way, Southampton, SO14 3ZH
[3]NORCE Norwegian Research Centre, Bjerknes Centre for Climate Research, Bergen, Norway

**Correspondence:** Charles Turner (charles.turner@soton.ac.uk)

**Abstract.** As the planet warms due to the accumulation of anthropogenic $CO_2$ in the atmosphere, the interaction of surface ocean carbonate chemistry and the radiative forcing of atmospheric $CO_2$ leads to the global ocean sequestering heat and carbon, in a ratio that is near constant in time. This ratio has been approximated as globally uniform, enabling the intimately linked patterns of ocean heat and carbon uptake to be derived. Patterns of ocean salinity also change as the earth system

warms due to hydrological cycle intensification and perturbations to air-sea freshwater fluxes. Local temperature and salinity change in the ocean may result from perturbed air-sea fluxes of heat and freshwater (excess temperature, salinity), or from reorganisation of the preindustrial temperature and salinity fields (redistributed temperature, salinity), which are largely due to circulation changes. Here, we present a novel method in which the redistribution of preindustrial carbon is diagnosed, and the redistribution of temperature and salinity estimated using only local spatial information. We demonstrate this technique in

the NEMO OGCM coupled to the MEDUSA-2 Biogeochemistry model under a RCP8.5 scenario over 1860-2099. The excess changes (difference between total and redistributed property changes) are thus calculated. We demonstrate that a global ratio between excess heat and temperature is largely appropriately regionally with key regional differences consistent with reduced efficiency in the transport of carbon through the mixed layer base at high latitudes. On centennial timescales, excess heat increases everywhere, with the North Atlantic a key site of excess heat uptake over the 21[st] century, accounting for 25% of the

total. Excess salinity meanwhile increases in the Atlantic but is generally negative in other basins, consistent with increasing atmospheric transport of freshwater out of the Atlantic. In the North Atlantic, changes in the inventory of excess salinity are detectable in the late 19[th] century, whereas increases in the inventory of excess heat does not become significant until the early 21[st] century. This is consistent with previous studies which find salinification of the Subtropical North Atlantic to be an early fingerprint of anthropogenic climate change.

Over the full simulation, we also find the imprint of AMOC slowdown through significant redistribution of heat away from the North Atlantic, and of salinity to the South Atlantic. Globally, temperature change at 2000m is accounted for both by redistributed and excess heat, but for salinity the excess component accounts for the majority of changes at the surface and at depth. This indicates that the circulation variability contributes significantly less to changes in ocean salinity than to heat content.

By the end of the simulation excess heat is the largest contribution to density change and steric sea level rise, while excess salinity greatly reduces spatial variability in steric sea level rise through density compensation of excess temperature patterns, particularly in the Atlantic. In the Atlantic, redistribution of the preindustrial heat and salinity fields also produce generally compensating changes in sea level, though this compensation is less clear elsewhere.

The regional strength of excess heat and salinity signals grows through the model run in response to the evolving forcing. In addition, the regional strength of the redistributed temperature and salinity signals also grow, indicating increasing circulation variability or systematic circulation change on timescales of at least the model run.

## 1 Introduction

As a result of continuing anthropogenic $CO_2$ emissions, atmospheric $CO_2$ levels continue to increase, as well as global mean surface air temperatures. However, the ocean acts to mitigate both changes, having absorbed a third of all $CO_2$ emissions to date (Khatiwala et al., 2013), as well as over 93% of the additional heat accumulating in the Earth system (Church et al., 2011). Though this greatly slows the rate of surface warming, it is not without consequence: as a result of the excess heat content, global sea levels are expected to rise significantly over the coming centuries, in large part due to the thermal expansion of seawater (thermosteric sea level rise) (Pardaens et al., 2011), (Church et al., 2013), with enormous implications for future economic development (Hinkel et al., 2014). It also has important implications for the future of marine ecosystem health: ocean warming has a direct effect on marine life as a driver of deoxygenation (Oschlies et al., 2018), as well as through increased stratification (Gruber, 2011). The uptake of carbon similarly affects marine life through its role in ocean acidification (Gruber, 2011).

As a result of the interaction of ocean biogeochemistry with rising atmospheric $CO_2$ and the increased radiative forcing it generates, there exists a near linear relationship between global mean surface air temperature change and cumulative carbon emissions, known as the Transient Climate Response to cumulative Carbon Emissions (TCRE) (Goodwin et al., 2015), (Katavouta et al., 2018). A similar near-linear global relationship exists between increases in ocean heat and carbon content (Bronselaer and Zanna, 2020), which can be observed at a range of scales: both increases in global ocean heat and carbon inventories, and in local ocean excess temperature and anthropogenic carbon are linearly related.

Local ocean heat content changes are contributed to by both the addition or removal of heat from the surface due to perturbed radiative forcing (excess heat), or from the rearrangement of the preindustrial temperature field from circulation variability (redistributed heat). Ocean salinity changes can also result from perturbations to air-sea freshwater fluxes (excess salinity), or due to the rearrangement of the preindustrial salinity field (redistributed salinity).

The redistribution of temperature and salinity as a result of ocean circulation variability acts on much shorter timescales than the accumulation of excess heat and salinity. Circulation-related variability comprises the majority of temporal variability in contemporary ocean temperature and salinity, (Bindoff and Mcdougall, 1994), (Desbruyères et al., 2017) and regional sea level (Church et al., 2013). However, the excess component is anticipated to dominate in the future (Bronselaer and Zanna

(2020), Zika et al. (2021)). Thus the evolution of excess temperature and patterns of excess salinity as well as changes in ocean circulation comprise a key source of uncertainty in estimates of regional sea level rise (Church et al., 2013).

While it remains challenging to separate excess and redistributed (preindustrial) heat, a similar decomposition for carbon is widely used. Identifying whether changes in ocean dissolved inorganic carbon (DIC) content are due to increased atmospheric $CO_2$ or changes in other processes (circulation, biological change, etc.) is possible due to the fact that atmospheric $CO_2$ can considered to be globally uniform, and biogeochemically-driven DIC changes may be parameterised. This allows us to separate changes in DIC into changes in anthropogenic carbon ($C_{anth}$, the DIC content considered to be due to increased atmospheric $CO_2$), and changes in natural carbon ($C_{nat}$, defined to be the non-$C_{anth}$ part of DIC) (Gruber et al. (1996), Hall et al. (2002), Touratier and Goyet (2004), Khatiwala et al. (2005), Vázquez-Rodríguez et al. (2009)). Natural carbon is therefore the sum of the pools of saturation carbon, carbonate carbon, soft tissue carbon and disequilibrium carbon: it can be thought of as the DIC field which exists in the ocean, prior to the Industrial Revolution (Williams and Follows (2011)), McKinley et al. (2017), Couldrey et al. (2019)). Although not precisely analogous (changes in natural carbon are not constrained to globally integrate to zero as with redistributed carbon), the decomposition of DIC into natural and anthropogenic components can provide valuable insights into excess and redistributed carbon (see Williams et al. (2021), Winton et al. (2013), Equation 6). Unlike carbon however, it is not straightforward to separate anthropogenically-driven changes in ocean temperature or salinity, due to the non-globally uniform nature of the perturbations: this has motivated a variety of techniques which aim to decompose excess and redistributed heat content changes.

One approach to determine excess temperature is to use a Passive Anomalous Tracer (PAT), which obeys the same physics as temperature, but is defined to have a preindustrial field which is zero everywhere: the preindustrial field therefore cannot contribute to redistribution (Banks and Gregory (2006), Gregory et al. (2016)), and so PAT reveals the distribution and evolution of the excess temperature field. Alternatively, it is possible in simulations to force ocean circulation to obey preindustrial dynamics despite increasing radiative forcing: this gives a similar result, though differing by a second order term to the PAT implementation (Winton et al., 2013).

While these methods have been very informative they are only applicable to models: no real world PAT tracer exists and while transient tracers such as chlorofluorocarbons are a close analogue their interpretation in terms of excess temperature necessitates the determination of an excess temperature boundary condition. This motivates the development of proxy methods, which aim to diagnose the excess and redistributed temperature from other tracers and might be more generally applied. The approach of Bronselaer and Zanna (2020) is an example of this: by approximating the distribution of excess temperature with that of anthropogenic carbon, they are able to leverage the mechanistic coupling relating excess heat accumulation to anthropogenic carbon accumulation to produce estimates of the scale and patterns of excess heat uptake.

Using an alternative carbon based methodology, Williams et al. (2021) explains differences in storage of heat and carbon in terms of two components: 1) the correlation of excess heat and carbon (both increase over time), and 2) anticorrelation of redistributed heat and carbon (the preindustrial distributions of temperature and carbon are inverted due to the inverse temperature dependence of carbon dioxide solubility). They use this to diagnose excess and redistributed heat (note Williams et al. (2021) refer to this as added heat, though the definitions used are identical). Bronselaer and Zanna (2020) can therefore

be thought of as specifying the character of this positive correlation between excess heat and anthropogenic carbon, in order to estimate excess heat directly from anthropogenic carbon. Here, we introduce an approach which builds on these ideas: we specify the character of the anticorrelation between redistributed heat and natural carbon locally via the preindustrial ocean state. This allows us to estimate redistributed heat (and other parameters) directly from redistributed carbon, which we may approximate using natural carbon. As natural carbon is strongly anticorrelated with temperature throughout the ocean, and can be usefully assumed to change only due to redistribution, it is an ideal tracer with which to estimate temperature redistribution.

Specifying the character of the relationship between the excess components of temperature and DIC change, as done by Bronselaer and Zanna (2020), relies on a global biogeochemical relationship derived from the radiative forcing of $CO_2$ and the ocean carbon buffer factor, making their approach applicable only to temperature. In contrast, in the absence of perturbations to mixing, redistribution leaves the properties of a parcel of water unchanged. As a result, the redistribution first approach we apply is more generally applicable: for example, if we identify a clear spatial relationship between natural carbon and salinity, we may use the redistribution of natural carbon to estimate the redistribution of salinity. This allows us to not only produce estimates of temperature redistribution, but also estimates of salinity, and by extension density, redistribution. Using these, we investigate the patterns of storage of excess and redistributed temperature and salinity by the global ocean.

## 2 Data and Methods

### 2.1 Model set up

We use the NEMO v3.2 OGCM (Ocean General Circulation Model) (Madec, 2008) coupled to the MEDUSA-2 biogeochemical model (Yool et al., 2013) and the Louvain-la-Neuve (LIM2) dynamic sea ice model (Timmermann et al., 2005). The model was configured with the ORCA1 grid with a nominal 1 degree resolution and 64 vertical levels (Madec and Imbard, 1996). The model was spun up for 900 years, before three 240 year simulations spanning 1860-2099 were spawned: a control run (CTR), coupled climate change run (COU), and a 'warming only' run (RAD), following the convention of Schwinger et al. (2014), Rodgers et al. (2020). The ocean model was forced with output from the HadGEM2-ES (Collins et al., 2011), an earth system model driven using prescribed greenhouse gas, land use and atmospheric chemistry forcing following the RCP8.5 scenario over the 1860-2099 time period. In this scenario, atmospheric $CO_2$ increases to over 900ppm by the end of the simulations (Riahi et al. (2011), atmospheric $CO_2$ in these simulations is shown in Couldrey et al. (2016), Figure 1a). Surface heat, momentum, freshwater fluxes, and atmospheric chemistry from HadGEM2-ES were used to force NEMO at 6 hourly intervals, and no restoring was used.

The CTR run is forced with 8 repetitions of the first 30 years of these fluxes from the HadGEM2-ES forcing, with a fixed atmospheric $CO_2$ of 286ppm: no significant climate change occurs in these 30 years. The 900 year spinup for all 3 model runs was also forced using this 30 year repeat forcing.

The COU run is forced with the full 240 year output from HadGEM2-ES. The RAD run has the same physical variability as in COU including that driven by atmospheric carbon increases but the atmospheric carbon is artificially relaxed to preindustrial conditions. As the RAD run only includes changes in DIC due to physical change (circulation change and warming), rather

than the ocean biogeochemical response to increased atmospheric $CO_2$, we can calculate this response, namely anthropogenic carbon or $C_{anth}$, directly from the difference of the COU and RAD runs:

$$C_{anth}(x,y,z,t) = DIC^{COU}(x,y,z,t) - DIC^{RAD}(x,y,z,t). \tag{1}$$

Natural carbon, or $C_{nat}$, is then defined to be the total DIC content with the anthropogenic carbon contribution removed: it is therefore calculated as

$$C_{nat} = DIC^{COU} - C_{anth} - \Delta DIC^{CTR} = DIC^{RAD} - \Delta DIC^{CTR}, \tag{2}$$

where $\Delta DIC^{CTR}$ is control run drift, equivalent to $DIC^{CTR}(x,y,z,t) - DIC^{CTR}(x,y,z,t_0)$, where $t_0$ is the beginning of the three simulations, 1860, and $\Delta$ refers to change since 1860 and $t$ is an arbitrary time. Therefore by definition all DIC is natural carbon at the beginning of our simulations, as the DIC fields are identical at the beginning of all 3 runs. DIC changes are then the sum of natural and anthropogenic carbon change. As such, we decompose the local DIC content at any time in the following way (note as $C_{anth}$ is defined to be zero at time $t = t_0$, $C_{anth} = \Delta C_{anth}$ here):

$$DIC(x,y,z,t) = DIC(x,y,z,t_0) + \Delta C_{nat}(x,y,z,t) + C_{anth}(x,y,z,t). \tag{3}$$

Changes in natural carbon, $\Delta C_{nat}$, are thus given by the difference in DIC between the RAD and CTR runs:

$$\Delta C_{nat} = DIC^{RAD} - DIC^{CTR}. \tag{4}$$

For further detail on model setup, see Couldrey et al. (2016) and Couldrey et al. (2019): we utilise the same simulations as these papers. We also note that Couldrey et al. (2019) compared the representation of DIC and alkalinity in these models runs to GLODAPv2 observations (Lauvset et al., 2016), finding the modelled carbon cycle to be representative of observations, and so we expect our carbon derived identification of excess temperature and salinity to also be representative.

## 2.2 Relating the redistribution of temperature and carbon

Following Williams et al. (2021), the preindustrial temperature and carbon fields of the ocean are broadly anticorrelated as a result of the strong inverse temperature dependence of carbon solubility. In contrast, the excess temperature and anthropogenic carbon fields are correlated due to the radiative forcing of atmospheric $CO_2$. Bronselaer and Zanna (2020) specify this correlation between excess heat and anthropogenic carbon using a time varying, globally uniform constant, which they refer to as the carbon-heat coupling or $\alpha$. Here, we aim to similarly relate the redistribution of temperature and natural carbon using an analogous redistribution coefficient, which we will label $\kappa_r$. As we also decompose salinity, we will use superscripts to denote the variable we are relating to natural carbon: the temperature redistribution coefficient, $\kappa_r^T$, refers to the preindustrial spatial covariability of natural carbon and temperature, whereas the salinity redistribution coefficient, $\kappa_r^S$ refers to the preindustrial spatial covariability of natural carbon and salinity. Decomposing the total temperature change,

$$\Delta\theta(x,y,z,t) = \Delta\theta_e(x,y,z,t) + \Delta\theta_r(x,y,z,t), \tag{5}$$

where $\theta$ is in situ potential temperature, and the subscripts $e$ and $r$ refer to the excess and redistributed components, respectively. We follow the definitions of Winton et al. (2013) for the excess and redistributed temperature:

$$\mathbf{v}\theta = (\mathbf{v}_0 + \mathbf{v}')(\theta_0 + \theta') = \underbrace{\mathbf{v}_0\theta_0}_{Preindustrial} + \underbrace{\mathbf{v}'\theta_0}_{Redistributed} + \underbrace{\mathbf{v}_o\theta' + \mathbf{v}'\theta'}_{Excess}, \tag{6}$$

where $\mathbf{v}_o$ and $\theta_0$ refer to the preindustrial components of the velocity field, $\mathbf{v}$, and the temperature field, $\theta$, and $\mathbf{v}'$ and $\theta'$ the perturbations. Salinity, DIC and $C_{anth}$ (or indeed any other tracer) changes may be decomposed in the same fashion. The excess component of a tracer can therefore be interpreted as changes in a tracer due to changes in surface forcing, and the redistributed component as changes in a tracer resulting from circulation change: redistribution driven changes in a tracer should therefore globally integrate to zero. At this point, we note that the preindustrial distribution of $C_{anth}$ is defined to be zero everywhere: thus $C_{anth}$ well approximates excess carbon. However, as further discussed later, $C_{nat}$ changes are not constrained to globally integrate to zero, and thus $C_{anth}$ and excess carbon, though similar, are not the same entity.

The approach of Bronselaer and Zanna (2020) is therefore to parameterise $\Delta\theta_e$ as

$$\Delta\theta_e(x,y,z,t) = \alpha_T(\Delta t) \times C_{anth}(x,y,z,t), \tag{7}$$

where $\alpha_T$ is their coefficient $\alpha$, expressed in units of temperature rather than heat. $\alpha$ is estimated as the ratio of global heat to DIC accumulation, over the time period $\Delta t = t - t_0$, where $t_0$ is a preindustrial time (1860 here). Alternatively, we might parameterise the redistribution of temperature, $\Delta\theta_r$ in terms of the natural carbon change:

$$\Delta\theta_r(x,y,z,t) \approx \kappa_r^T(x,y,z) \times \Delta C_{nat}(x,y,z,t). \tag{8}$$

Unlike $\alpha_T$, $\kappa_r^T$ is not a function of time: it is instead a function of position, as it relates the spatial covariability of the preindustrial temperature and carbon fields at a given point. This method is equally applicable to any property for which we aim to estimate redistribution although each property pair will have a distinct distribution of $\kappa_r$: we could instead choose to relate the spatial covariability of the preindustrial salinity and carbon fields. We would therefore estimate redistributed salinity, $\Delta S_r$, as

$$\Delta S_r(x,y,z,t) \approx \kappa_r^S(x,y,z) \times \Delta C_{nat}(x,y,z,t). \tag{9}$$

In Equations 8 and 9, no constraint is made such that the global integral of redistributed heat is zero (or equivalently the global mean redistributed temperature is zero). If $C_{nat}$ were a perfect tracer for redistribution, then its global integral would be zero. However, we expect the global integral of $\Delta C_{nat}$ to be nonzero, predominantly as a result of the outgassing of saturation carbon, $C_{sat}$ (the DIC content of the ocean resulting from equilibrium with the preindustrial atmosphere), in response to ocean warming. Thus the quantities $\Delta DIC_r$ (redistributed DIC) and $\Delta C_{nat}$ will differ, particularly over timescales of multiple decades to centuries (Williams et al., 2021): to reflect this, we have used approximate rather than exact equality in Equations 8 and 9. In general, when integrating over the global ocean,

$$\frac{d}{dt}\iiint C_{nat}\,dV < 0, \tag{10}$$

so we correct for the divergence of $C_{nat}$ and the ideal behaviour of a redistributed preindustrial carbon field using a repartitioning factor, which we refer to as $\gamma$. We refer to the corrected quantity as adjusted natural carbon, $C_{nat}^{adj}$. We use this to repartition a fraction of anthropogenic carbon into the adjusted natural carbon in order to correct for $C_{sat}$ outgassing.

This repartitioning allows us to force the global integral of adjusted natural carbon changes to zero. However, because globally integrated biology driven changes in $C_{nat}$ may be nonzero, we instead enforce the condition that globally integrated redistributed heat, not adjusted natural carbon, is zero. We therefore estimate the redistributed temperature field as

$$\Delta\theta_r(x,y,z,t) = \kappa_r^T(x,y,z) \times \Delta C_{nat}^{adj}(x,y,z,t) = \kappa_r^T(x,y,z) \times \Big(\Delta C_{nat}(x,y,z,t) + \gamma(t)C_{anth}(x,y,z,t)\Big), \tag{11}$$

where $\gamma(t)$ is a factor between 0 and 1 such that over the global ocean

$$\iiint \Delta\theta_r dV = 0 \tag{12}$$

at all times. $\gamma$ must be less than 1 or $C_{nat}^{adj}$ would exceed DIC, and so would not be physically meaningful. It is constrained to be positive as historically, atmospheric $CO_2$ has increased from preindustrial levels and so the global $C_{anth}$ inventory is positive. However, if the global $C_{anth}$ inventory were negative, $\gamma$ could also be negative (though the magnitude is always less than 1).

As with redistributed temperature, we will estimate redistributed salinity as

$$\Delta S_r(x,y,z,t) = \kappa_r^S(x,y,z) \times \Delta C_{nat}^{adj}(x,y,z,t) = \kappa_r^S(x,y,z) \times \Big(\Delta C_{nat}(x,y,z,t) + \gamma(t)C_{anth}(x,y,z,t)\Big). \tag{13}$$

We also note that we may combine Equations 11 and 13 in order to directly estimate salinity redistribution from temperature redistribution and vice versa. This follows from the property that (assuming no perturbation to mixing) redistribution does not alter the properties of a parcel of water, and so the redistribution of natural carbon, temperature and salinity are related by the spatial covariability of their preindustrial fields. Alternatively, we may observe that the choice of $C_{nat}$ is not unique as a tracer for which to estimate redistribution: as we previously note, we only a require a tracer which can be considered to change only through redistribution. The sum of the preindustrial temperature or salinity fields and their redistributed components both satisfy this, and so can be used to estimate redistribution of other tracers themselves.

We now wish to estimate our redistribution coefficients relating the change in adjusted natural carbon to changes in temperature ($\kappa_r^T$) and salinity ($\kappa_r^S$), in order to determine their redistribution. To do this, we use a statistical method, examining how the model temperature or salinity and $C_{nat}$ fields covary on subdecadal timescales in our control run, in order to estimate the covariability of their preindustrial state. It is well known that when making repeated observations at a fixed spatial location, the majority of observed changes in temperature and salinity are due to circulation variability, rather than material changes in water mass properties on subdecadal timescales (for example Bindoff and Mcdougall (1994), Firing et al. (2017)). We exploit the dominance of circulation variability on these timescales, assuming that the correlation between deviations in temperature, salinity and DIC from their mean state on subdecadal timescales are due entirely to circulation variability. The correlations obtained allow us to estimate how circulation acts to couple changes in temperature and salinity to changes in natural carbon, at every point in the ocean. Thus, by looking at the relationship between temperature or salinity and DIC on subdecadal

timescales, we are able to identify the spatial covariability of the background fields, without an explicitly decomposed temperature or salinity field. Though these spatial correlations will change due to the addition of excess temperature (salinity), excess and redistributed temperature (salinity) are defined such that these preindustrial correlations correctly capture the relationship between redistributed temperature (salinity) and natural carbon change throughout the COU simulation (Equation (6)).

     The calculation is performed as follows: in each grid cell, we use the full 240 years (1860-2099) of yearly mean temperature,

salinity and DIC from our CTR run, binned into 24 decades. In each decadal bin, the mean tracer ($\theta$ or $S$ and $C_{nat}$) values are subtracted, giving yearly $\theta$, $S$ and $C_{nat}$ anomalies from the decadal mean in that grid cell. We perform this decadal binning in order to preclude the possibility of any excess temperature or excess DIC contaminating our relationship as the result of models drifts or surface forcing driven variability due to the 30 year repeated forcing: though these effects should be small, they are both partitioned by the excess/redistribution decomposition into excess.

The correlations between the yearly anomalies from decadal means, for the entire 240 years of data, are then used to establish an intermediate value, which we label $\kappa_i$, at each grid cell. This is done using a total least squares linear fit, implemented as two dimensional PCA: we estimate $\kappa_i$ as the gradient of the slope obtained. We perform a total least squares fit, rather than an ordinary least squares fit, as we expect the two variables to be correlated, but not causally: total least squares is therefore more appropriate, as our relationship should not be affected by the choice of dependent variable.

We then calculate a suppression factor, $\phi$, based on the quality of the correlations to estimate $\kappa_r$ for each variable: this process is detailed in Appendix (A), along with a visualisation of the estimation process. As with $\kappa_r$, this will be unique to each variable. $\phi$ is designed such that where the correlations we obtain between the $\theta$ or $S$ and $C_{nat}$ anomalies from decadal means are poor or nonexistent, no estimate of redistribution is made. As a result of this, if local $C_{nat}$ changes due to biological processes but temperature or salinity due to circulation variability, our method will misclassify this as excess temperature

or salinity: this also will occur at maxima/minima of temperature or salinity. However, due to the implementation of our $\gamma$ correction, these misclassifications will globally integrate to zero. Over the full simulation, adjusted natural carbon increases by approximately $2\mu$mol/kg, 0.1% of the mean preindustrial DIC concentration. This implies the net global divergence of $C_{nat}^{adj}$ and $\Delta DIC_r$ is approximately 0.1%.

     The full calculation is therefore performed as

$$\Delta\theta_r(x,y,z,t) = \kappa_r^T(x,y,z) \times \Delta C_{nat}^{adj}(x,y,z,t) = \phi_\theta(x,y,z) \times \kappa_i^T(x,y,z) \times \Delta C_{nat}^{adj}(x,y,z,t) \tag{14}$$

for temperature, and

$$\Delta S_r(x,y,z,t) = \kappa_r^S(x,y,z) \times \Delta C_{nat}^{adj}(x,y,z,t) = \phi_S(x,y,z) \times \kappa_i^S(x,y,z) \times \Delta C_{nat}^{adj}(x,y,z,t) \tag{15}$$

for salinity. Excess temperature was then calculated as

$$\Delta\theta_e(x,y,z,t) = \Delta\theta(x,y,z,t) - \Delta\theta_r(x,y,z,t) = TMP^{COU}(x,y,z,t) - TMP^{CTR}(x,y,z,t) - \kappa_r^T(x,y,z) \times \Delta C_{nat}^{adj}(x,y,z,t),$$
$$\tag{16}$$

and likewise for salinity.

Estimates of redistributed salinity are complicated in the top 200m by the impacts of freshwater dilution, leading to misattribution of excess salinity to redistribution. To resolve this issue, we recalculate salinity redistribution using the same statistical approach to locally estimate the salinity redistribution from the redistributed temperature field: we refer to this as a two step estimation. This calculation is performed as

$$\Delta S_r^2(x,y,z,t) = \kappa_r^{T-S}(x,y,z) \times \Delta\theta_r(x,y,z,t) = \kappa_r^{T-S}(x,y,z) \times \phi_T(x,y,z) \times \kappa_i^T(x,y,z) \times \Delta C_{\text{nat}}^{\text{adj}}(x,y,z,t), \tag{17}$$

where the superscript 2 in $\Delta S_r^2(x,y,z,t)$ refers to the two step estimation. $\kappa_r^{T-S}$ is an estimate of the T-S curve angle, and is estimated in the same way as $\kappa_i^T$ and $\kappa_i^S$: we do not apply a new suppression factor.

The two redistributed salinity estimates were then combined using a sigmoidal weighting, exchanging from the two step estimate at the surface to the one step estimate at depth with equal weight at 200m. This was not found to leave any artefacts
in the estimates of salinity redistribution. This process is detailed in Appendix (B).

For temperature, approximately 80% of grid cells globally have a scale factor of 0.8-1, and we find by the end of our run, the suppression factor $\phi$ alters the redistributed temperature of 93% grid cells globally by less than 0.04 degrees, and 60% by less than 0.02 degrees, though the RMS mean redistributed temperature is reduced by 5%. However, the small number of grid cells producing extremely large estimates (10's of degrees of change) are effectively suppressed. We therefore estimate that the
statistical nature of our method introduces a minimum uncertainty of approximately 5% into our inventories.

$\gamma$ was calculated for each year using Equation (11) to satisfy Equation (12): a fraction of $C_{\text{anth}}$ was added to $C_{\text{nat}}$ to ensure the global integral of redistributed heat is zero in each year, with the fraction representing the $\gamma$ value that year. We then smooth the value of $\gamma$ over a 10 year period, before adding the fraction of $C_{\text{anth}}$ each year given by our smoothed $\gamma$ value to $C_{\text{nat}}$ to obtain our $C_{\text{nat}}^{\text{adj}}$ field. Over the 240 year run, $\gamma$ increases from 0 to approximately 0.12, with an approximately sigmoid shape.
This is shown in Appendix (C).

Once the $C_{\text{nat}}^{\text{adj}}$ field had been built, it was used to generate both the redistributed temperature and salinity fields: we did not recalculate a new $\gamma$ value to force a zero integral of salinity redistribution in our salinity decomposition. This approach was chosen for 3 reasons. Calculating a new $\gamma$ for salinity would mean a new $C_{\text{nat}}^{\text{adj}}$ field, and so the evolution of the redistributed temperature and salinity fields would not be linked by the same adjusted $C_{\text{nat}}^{\text{adj}}$ field. In addition, the salinity of sea ice in the
model (6PSU) and reduced carbon content of sea ice causes some ice melt to be captured as redistributed salinity, rather than excess. This means that we do not expect globally integrated salinity redistribution to sum to zero as we do for temperature. Finally, as globally integrated redistributed salinity is not independently constrained to be zero, this allows us to use this global integral as a check on the validity of the method.

Excess and redistributed density fields were then built from the decomposed temperature and salinity fields. To do this, the
redistributed fields were added to the initial fields, and redistributed density calculated using TEOS-10 (McDougall and Barker, 2011). Initial density was then subtracted for density redistribution. Excess density fields were then calculated as the difference between the redistributed density field and the total density change.

# 3   Results

## 3.1   Methodology Validation

We validate our results by comparison with previous carbon proxy based methods. The method of Bronselaer and Zanna (2020) relies on a globally uniform $\alpha$ value, linking carbon and heat changes at all scales, which they refer to as the carbon-heat coupling. In comparison, our technique does not enforce global uniformity of this carbon-heat coupling: a local carbon-heat coupling, $\Delta\theta_e/\Delta C_{\mathrm{anth}}$, is instead an output of our method. Henceforth, we will refer to the global mean carbon-heat coupling as $\alpha_T$, and the local carbon-heat coupling as $\Delta\theta_e/\Delta C_{\mathrm{anth}}$: specifically, the local carbon-heat coupling links the anthropogenic carbon and excess heat.

As we expect the correlations between the excess components of temperature and DIC changes to be positive, and between the redistributed components of temperature and DIC changes to be negative, we can infer whether our technique reliably estimates excess heat by comparing histograms of correlations between the different components of temperature and carbon change. To do this, we compare the total temperature change to DIC change, the excess temperature change to $C_{\mathrm{anth}}$ change, and the redistributed temperature change to $C_{\mathrm{nat}}^{\mathrm{adj}}$ change (equivalent to $\kappa_r^T$ in our method), for each grid cell at depths of less than 2000m in our simulations. We exclude depths greater than 2000m due to the negligible ventilation and $C_{\mathrm{anth}}$ beyond this depth horizon. The total change and excess component correlations are calculated as the ratio of decadal mean temperature and carbon at each grid cell for the period 2090-2099 minus the initial values in 1860. Assuming the assumption of a globally uniform $\alpha_T$ to be accurate, we expect to find a broad distribution of ratios of total temperature change to DIC change with both positive and negative correlations, and a narrower distribution of ratios of excess temperature change to $C_{\mathrm{anth}}$ change, centred about the global mean value $\alpha_T$. We also expect the correlations between redistributed temperature changes and $C_{\mathrm{nat}}^{\mathrm{adj}}$ to generally be negative.

This is shown in Figure 1: the volume weighted histogram of each of these quantities over the upper 2000m of the ocean. The distribution of the ratio of total temperature change to DIC change (black line) is generally positive, indicating the dominance of excess temperature and DIC over redistribution over this time period and region, but is broad and encompasses both positive and negative values. It's mode occurs at the global mean value $\alpha_T$: 0.016K/$\mu$mol/kg. The mode of the ratio of excess temperature change to $C_{\mathrm{anth}}$ accumulation (red line) is slightly lower (0.012-0.014K/$\mu$mol/kg), but the magnitude of the peak at the mode is approximately 50% greater than that of total change ($2.1 \times 10^{16}$m$^3$ vs $1.4 \times 10^{16}$m$^3$). This implies that the assumption of a globally uniform carbon-heat coupling, $\alpha$, is broadly appropriate although a large spread in values exists, and that our method reliably identifies excess heat.

The distribution of the ratio of redistributed temperature change to $C_{\mathrm{nat}}^{\mathrm{adj}}$ change ($\kappa_r^T$, blue line), is also generally negative, as expected, with a much broader distribution than the distribution of the ratio of excess temperature and $C_{\mathrm{anth}}$. Generally, the intermediate value histogram ($\kappa_i^T$, magenta line) resembles the final ratio ($\kappa_r^T$, blue line), with the exception of the large peak at zero, resulting from the suppression factor, $\phi_T$. The positive tail of $\kappa_r^T$ values is predominantly due to the inversion of the DIC field with depth in the North Pacific (see Figure D1, distributions of both $\kappa_r^T$ and $\kappa_r^S$ are shown in Appendix (D)). That the correlation between the redistribution of temperature and carbon is positive here implies that the method of Williams

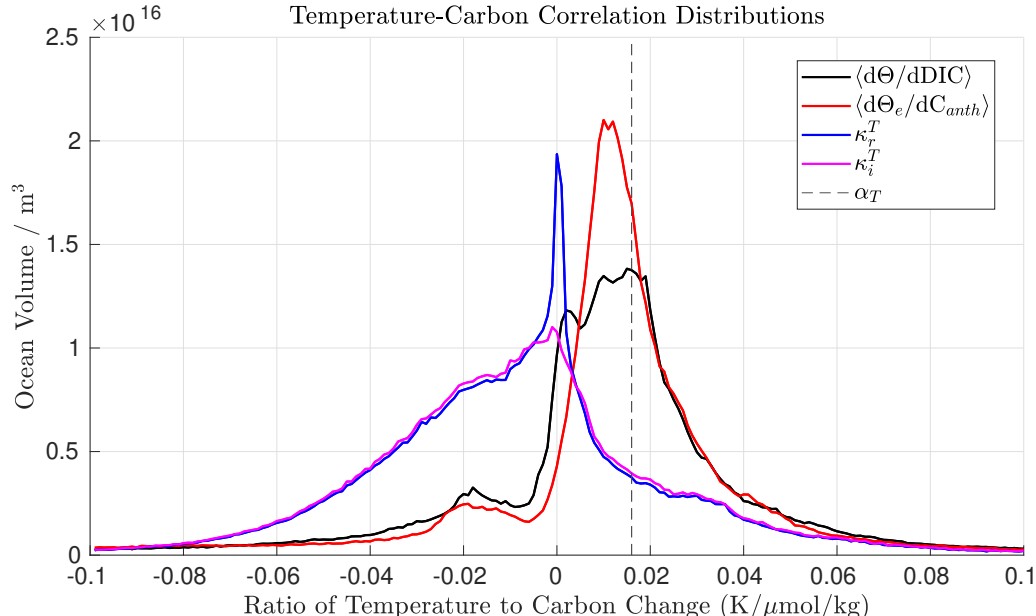

**Figure 1.** Histograms of the distribution of correlations relating different components of the temperature and carbon fields, over the full simulation (1860-2099). The global mean value $\alpha_T$ is shown by the dashed line. We include both the final redistribution coefficient, $\kappa_r^T$ (blue), and its intermediate estimate, $\kappa_i^T$ (magenta), as well as the ratio of total temperature change to DIC change ($\langle d\theta/d\mathrm{DIC}\rangle$, black), and local excess temperature to anthropogenic carbon change ($\langle d\theta_e/d\mathrm{C_{anth}}\rangle$, red).

et al. (2021) may not be appropriate in this location . However, the shape of our distributions are in clear agreement with their method: our decomposition generally infers negative correlations between redistributed temperature and natural carbon, and positive correlations between excess temperature and anthropogenic carbon. As our method identifies correlations between
excess temperature and anthropogenic carbon, and between redistributed temperature and natural carbon changes that are consistent with both the assumptions of Williams et al. (2021) and Bronselaer and Zanna (2020), despite not enforcing this to be the case, we have confidence that it is reliably separating excess and redistributed temperature.

We now compare estimates of excess temperature from our method and that of Bronselaer and Zanna (2020): both methods producing consistent estimates indicates we are accurately identifying the excess temperature field. Figure 2 shows the zonally
averaged excess and total temperature fields we obtain for the Atlantic and Indo-Pacific, for the final decade of our simulations, 2090-2099. In the Atlantic and Indo-Pacific, the estimate using the method of Bronselaer and Zanna (2020) (2a,b) produces smoother estimates than our technique (2c,d), but there are a number of common features which both techniques identify that are not due to the accumulation of excess heat. In the Atlantic, the tongue of warming at 2000-2500m depth, extending from approximately 40°N to 30°S is identified by both techniques as redistribution of the preindustrial temperature field, rather than
excess heat. In addition, both techniques identify the region of warming extending from approximately 2000-4000m depth between 60°S and 40°S as redistribution, rather than excess heat. In the Indo-Pacific, both methods identify the cooling at

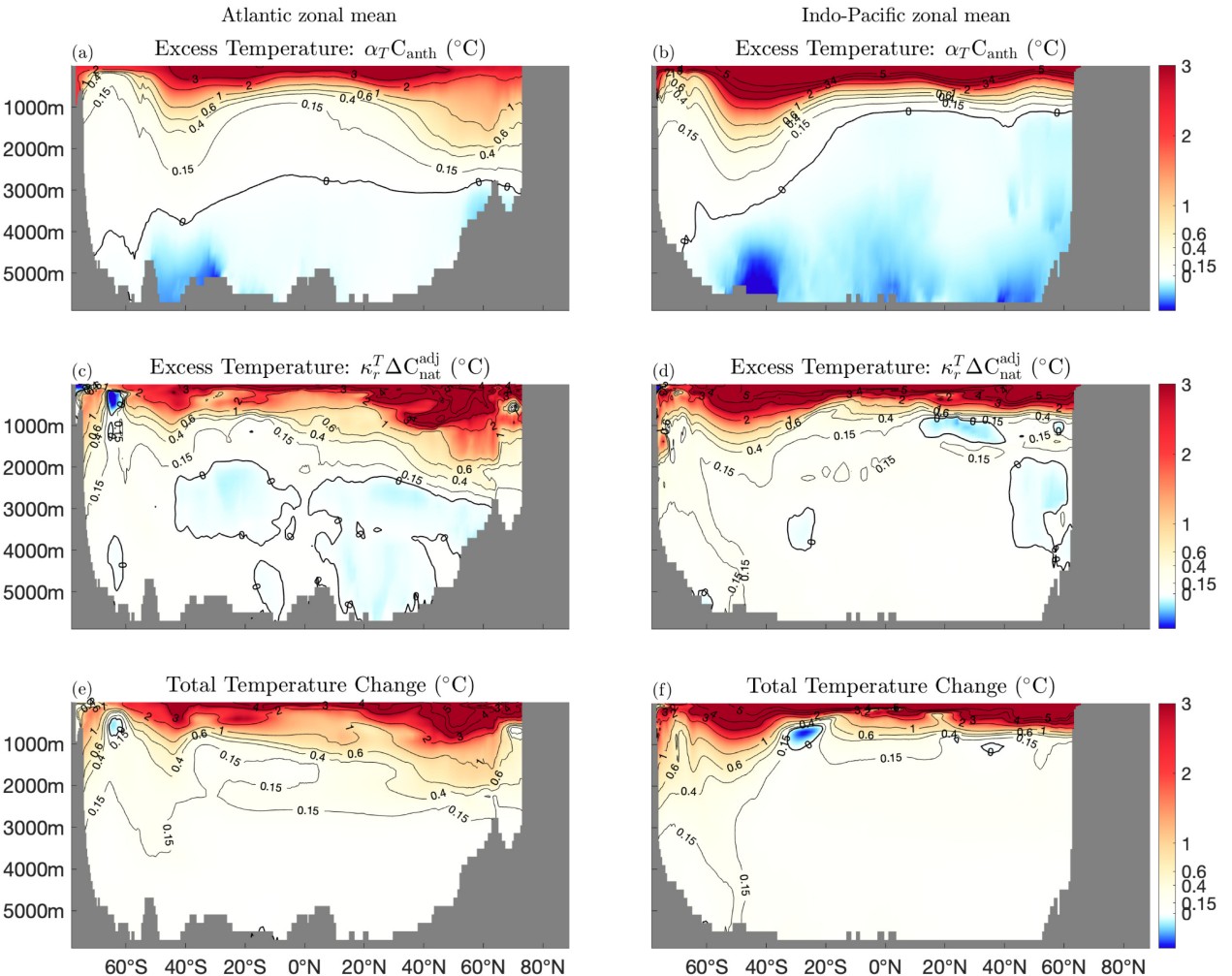

**Figure 2.** Atlantic and Indo-Pacific zonal, decadal mean excess temperature estimates, for the decade 2090-99, and total temperature change. The method of Bronselaer and Zanna (2020) is shown in panels (a,b), our method in panels (c,d), and the total temperature change in panels (e,f). The thick black contour indicates the zero contour, and temperature changes are indicated by thin contours, which are also indicated on the colour axes.

approximately 1000m at 20°S as redistribution, rather than excess temperature. However, our method identifies the penetration of excess temperature to depth in the Southern Ocean, unlike the method of Bronselaer and Zanna (2020).

In the upper 1000m, there are significant divergences between the two techniques. To explore the sources of these differences, we compute local estimates of the quantity $\Delta\theta_e/\Delta C_{\mathrm{anth}}$ from our estimates of $\Delta\theta_e$ and model $C_{\mathrm{anth}}$. By comparing our locally obtained estimates with the patterns of excess heat and anthropogenic carbon uptake estimated by assuming a globally uniform

$\alpha_T$, we are able to show how our relaxation of the assumption of a globally uniform $\alpha_T$ causes our estimates to differ. This is demonstrated in Figure 3.

Figure 3a and 3b show local values of $\Delta\theta_e/\Delta C_{\mathrm{anth}}$, presented as the zonal mean of the ratio of total excess temperature accumulated to total anthropogenic carbon accumulated, averaged over the decade 2090-2099. Figure 3c and 3d show the differences between the excess temperature estimated using our technique, and estimated using the technique of Bronselaer and Zanna (2020), and Figure 3e, 3f shows the total $C_{\mathrm{anth}}$ accumulated over the same period and domain.

At depths of below 2000m in the Atlantic and 1000m in the North Pacific, ventilation is negligible and so despite large $\Delta\theta_e/\Delta C_{\mathrm{anth}}$ estimates, the two methods produce similar estimates of excess temperature. In the Southern Ocean, North Atlantic and North Pacific, we see large $\Delta\theta_e/\Delta C_{\mathrm{anth}}$ values, as well as nontrivial accumulation of excess temperature. We therefore find that in these regions, our estimates of excess temperature and those using the method of Bronselaer and Zanna (2020) diverge.

In general, our estimates of $\Delta\theta_e/\Delta C_{\mathrm{anth}}$ show a large degree of spatial coherence, despite no constraints being imposed to enforce this. This gives us confidence that these variations are likely real, rather than an artifact of our estimation technique. An implication of this is that heat uptake is intensified, relative to $C_{\mathrm{anth}}$ uptake, in the high latitude Northern Hemisphere, and reduced in the low latitude Northern Hemisphere and Southern Hemisphere. We suggest that this may be explained in terms of a reduction of carbon export through the mixed layer at high latitudes. Bronselaer and Zanna (2020) make an argument for a globally uniform $\alpha$ value based on surface carbonate chemistry. However, Bopp et al. (2015) found total $C_{\mathrm{anth}}$ subduction through the base of the mixed layer to be significantly more variable than air-sea $C_{\mathrm{anth}}$ fluxes, and generally reduced at high latitudes (their Figure 3c): this mechanism could potentially act to reduce the spatial uniformity of $\alpha$ below the base of the mixed layer. In particular, water masses where the effects of advection and vertical mixing on carbon subduction are in opposition (namely high latitudes) tend to produce higher values of $\Delta\theta_e/\Delta C_{\mathrm{anth}}$.

To test whether these variations in local values of $\Delta\theta_e/\Delta C_{\mathrm{anth}}$ may constitute a source of error in the method of Bronselaer and Zanna (2020), we also compare the column inventories of excess heat uptake over the top 2000m of the ocean obtained using both methods in our simulations: this is shown in Figure 4. Bronselaer and Zanna (2020) were able to directly compare their estimates of excess heat and the simulated excess heat (their Figure 3f). We find that though our estimates do differ, these differences (Figure 4c) closely resemble those between their method and the simulated excess (their 3f). The zonally integrated difference in upper 2000m excess heat content (Figure 4d) is again consistent with a reduction of carbon export through the mixed layer base at high latitudes.

As our method produces results broadly consistent with both the method of Bronselaer and Zanna (2020) and Williams et al. (2021), we believe it is reliably identifying excess temperature. In addition we find a plausible explanation for differences between the results of the two methods, that is consistent with the inference that the spatial variability in the ratio of $C_{\mathrm{anth}}$ and excess heat accumulation is realistic. In Appendix E, we explore the accuracy of our decomposition in regions of the ocean which can be assumed to be unventilated during the simulations. These results suggest that our method reasonably captures the higher frequency (subdecadal timescales) variability in ocean heat content due to circulation variability, and captures at least

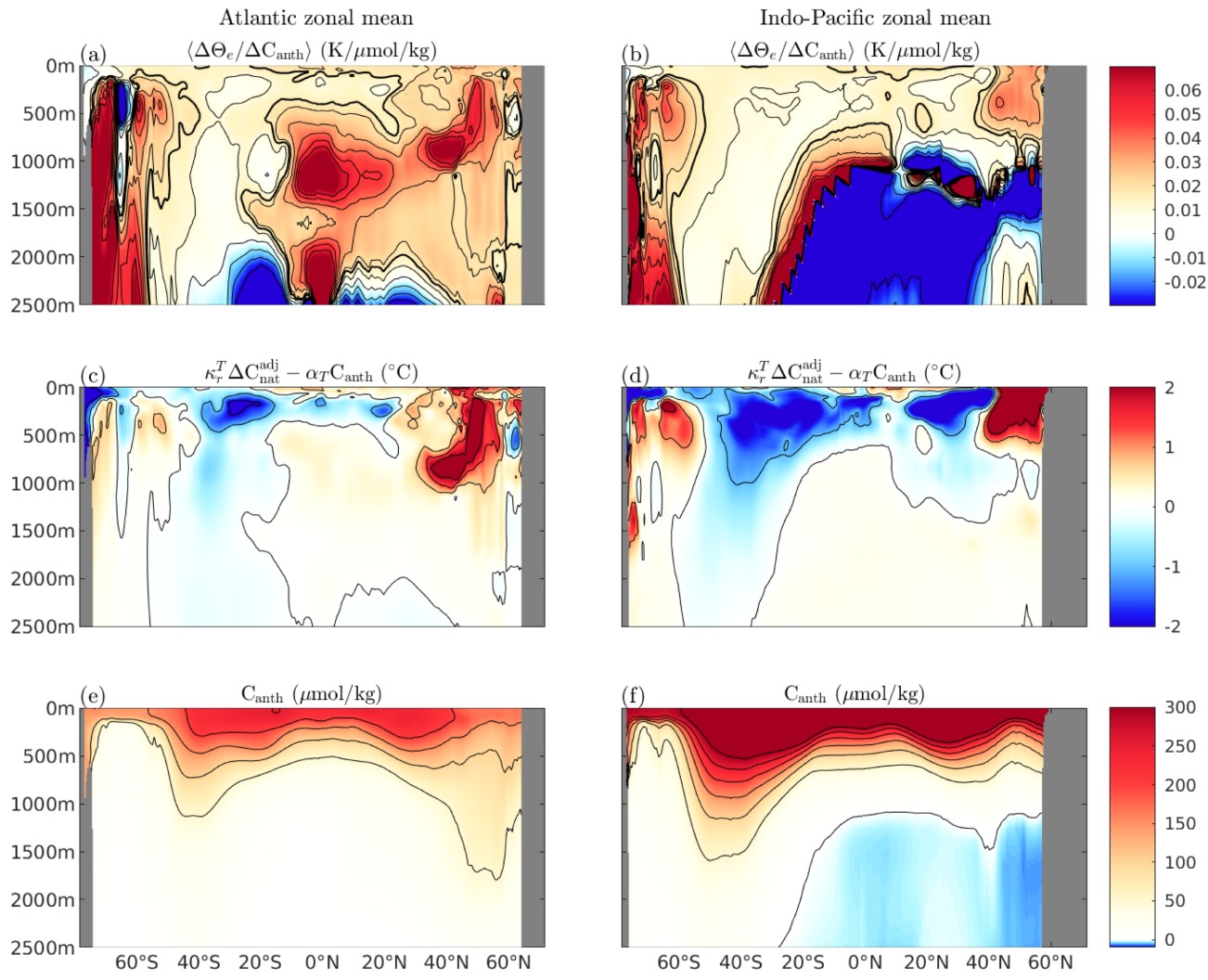

**Figure 3.** Atlantic and Indo-Pacific zonal mean ratio of excess temperature to $C_{anth}$ accumulation, calculated as the 2090's decadal, zonal mean temperature divided by 2090's decadal, zonal mean $C_{anth}$ (Panels (a), (b)). Panels (c) and (d) show the difference between our excess temperature estimate and the excess temperature estimate produced using the method of Bronselaer and Zanna (2020), and Panels (e) and (f) the zonal mean $C_{anth}$ accumulation, calculated as the 2090's decadal mean. The thick black contour in Panels (a), (b) indicate the global mean value of $\alpha_T$ of 0.016K/$\mu$mol/kg, and the thin contours are indicated on the colour axes.

80% of the longer timescale (centennial) redistributed heat content. The uncertainty of redistributed heat inventories introduced by our method is thus calculated as being in the range of 5-20%.

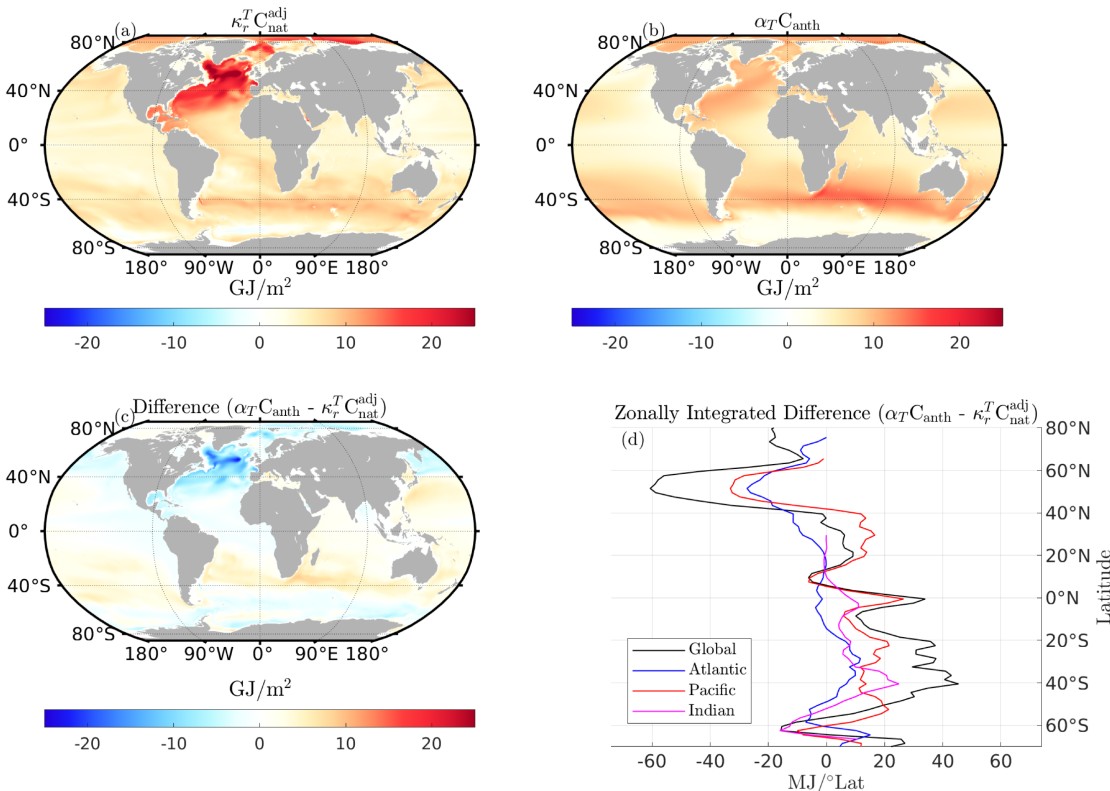

**Figure 4.** Column inventories of excess heat (0-2000m), calculated as the 2090-2099 decadal mean excess temperature relative to prein-dustrial. Panel (a) shows our method, panel (b) the method of Bronselaer and Zanna (2020), and panel (c) the difference between the two estimates. Panel (d) shows the zonally integrated upper 2000m excess heat content difference.

## 3.2 Inventory Changes

Global mean excess and redistributed salinity change, as well as globally integrated excess and redistributed heat content

change are shown in Figure 5. The global mean excess and redistributed salinity (thick lines, Figure 5a) begin to decrease in 1891, when the RAD and CTR forcing ceases to be identical, though this sea ice melt driven decrease is much smaller than the scale of either the positive and negative only excess or redistributed salinity components (thin dashed lines): global mean excess and redistributed salinity both decrease by approximately 0.001PSU over the full run. Globally integrated excess heat does not begin to accumulate significantly until approximately 2000: until this point, both positive only (global integral of

excess heat content only in regions where excess temperature is positive) and negative only excess and redistributed heat are of similar scales. Positive only excess heat and globally integrated excess heat are approximately the same by 2050, and negative only excess heat increases from approximately $-200$ZJ in 2000 to approximately $-50$ZJ by 2100: some negative excess heat due to cooling in the first half of the run remains throughout the full simulation.

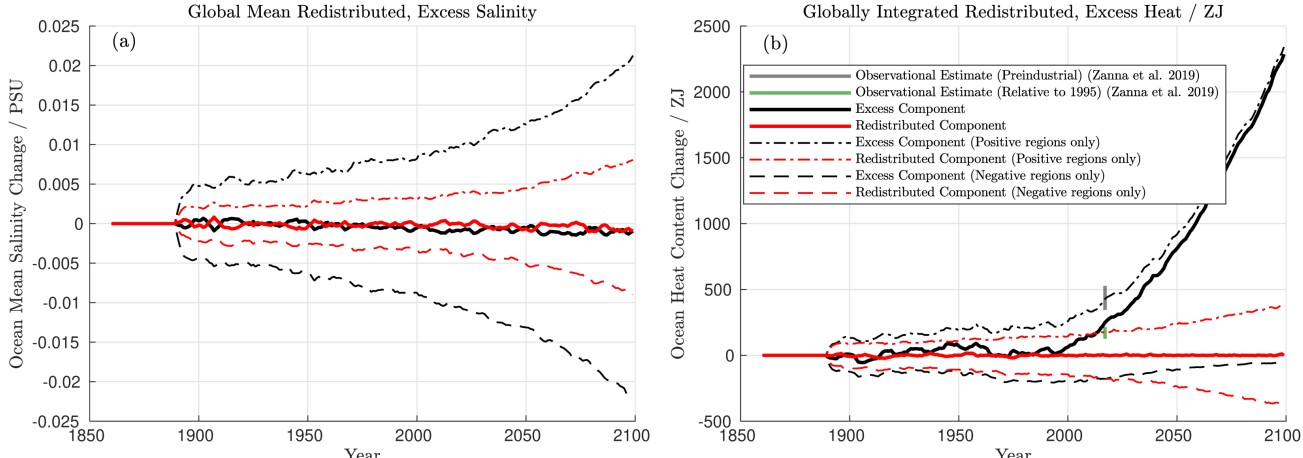

**Figure 5.** Global mean excess and redistributed salinity (a), and globally integrated excess and redistributed heat (b). Excess components are shown in black, redistributed components in red. The integrals of only the positive and negative regions are also shown (thin dashed lines). Climate change and control runs use the same first 30 years of forcing, so values are by definition zero here: the jump in 1890 represents the initial divergence of states. Observational estimates of global ocean heat uptake from Zanna et al. (2019) are also shown in panel (b).

The global integral of positive and negative only regions are useful for assessing the extent of redistribution: whilst the
global integral of redistributed temperature is constrained to be zero, this is the result of the cancellation of the positive and negative regions. Whilst excess heat begins to dominate during the mid 21$^{st}$ century, the extent of temperature (and salinity) redistribution continually increases: there is no indication of 'settling' into a new circulation state, where redistribution ceases to increase, on the timescale of the full simulation. This can be seen from the continued and accelerating increases in positive and negative only redistributed heat and salinity. We also observe that whilst the magnitude of positive and negative only
redistributed heat are similar until approximately 2000, excess salinity is significantly larger than redistributed salinity at all times. This indicates that during the full course of our simulations, salinity changes are dominated by changes in the freshwater cycle, rather than changes in circulation.

For comparison, we include observational estimates of ocean heat uptake from Zanna et al. (2019) (Figure 5b): cumulative heat uptake over 1871-2015 in grey (436±91ZJ) and over 1995-2015 in green (153±44ZJ). Over the period 1871-2015, our
simulations underestimate cumulative heat uptake (249ZJ), but overestimate heat uptake over 1992-2015 (232ZJ).

Figure 6 shows the integrated redistributed and excess temperature, salinity, and densities for each ocean basin. As with the global mean, excess salinity begins to accumulate almost immediately in most ocean basins (Figure 6c), particularly the North Atlantic and South Pacific: trends here are distinct from noise at $2\sigma$ in 1893 and 1911, respectively. Excess temperature does not begin to accumulate until the 21$^{st}$ century, at which point it begins to rapidly accumulate in all ocean basins; the exception
to this is the South Atlantic (Figure 2a, dashed black line) which cools in the 20$^{th}$ century, its excess heat signal emerging from noise at $2\sigma$ in 1918. In contrast, the excess heat signal in the North Atlantic and South Pacific do not emerge from noise at $2\sigma$

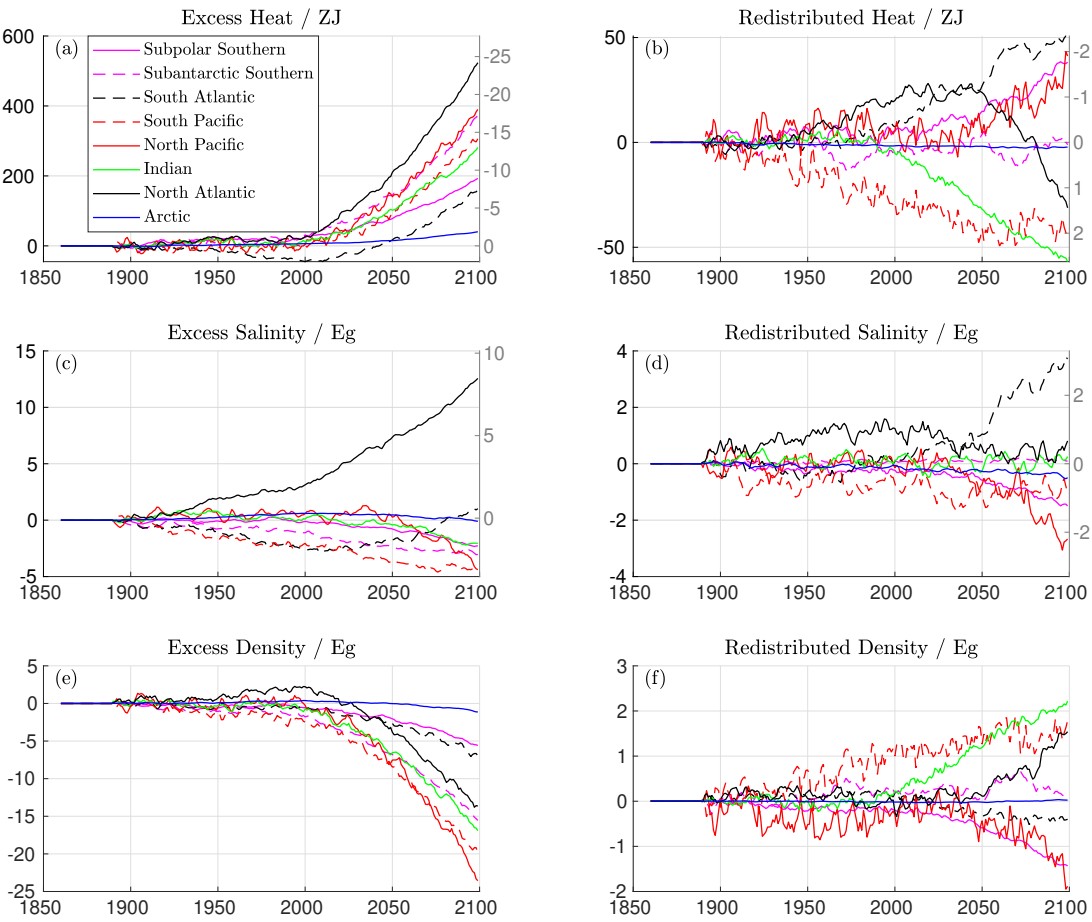

**Figure 6.** Excess (left column) and redistributed (right) heat, salinity and density integrals for each ocean basin over the full model run. For the changes in heat and salinity (Panels (a)-(d)), the equivalent integrated density change in units of Pg are given in grey on the right. Scales differ for excess and redistributed components, and changes in salinity and density are given as mass changes rather than volumes.

until 2023 and 2021, respectively. Over the period 2023-2099, for which the excess heat signal of the North Atlantic is distinct from noise, 25±2% of global excess heat accumulated is located in the North Atlantic.

The accumulation of negative excess density is dominated by the accumulation of excess temperature, rather than salinity: the grey scales on the right hand side of panels (a)-(d) show the density change associated with heat and salinity change. In the North Atlantic, changes in the excess heat and salinity compensate to reduce density anomalies: a reduction of almost 25Pg associated with excess heat is compensated for by an increase of approximately 8Pg associated with increased salinity. Similar compensation, though much weaker, is seen in the South Atlantic, which cools and freshens during the 20<sup>th</sup> century before

warming and salinifying in the 21st. This is not the case in other basins, where the changes in excess heat and salinity both act
to decrease density and therefore increase stratification.

The redistribution of density is less dominated by heat, with heat and salinity contributing similarly to redistributed density. In the North Atlantic, the redistribution of heat and salinity are approximately density compensated until around 2050, at which point the redistributed density inventory begins to increase rapidly (Figure 6f, black line). Good density compensation in the redistributed component is also seen in the subantarctic Southern Ocean, with minimal accumulation of redistributed density.

In our COU run, AMOC strength (calculated as peak depth integrated meridional volume transport at 26°N) increases until 1990 before declining continually thereafter. The cumulative transport anomaly (time integrated difference between COU and CTR AMOC volume transport) peaks in 2035 before also declining continually for the rest of the simulation. The signal of AMOC decline is visible in the redistributed heat content of the North Atlantic, which peaks in 2037 before declining rapidly, as well as the redistributed salinity content of the South Atlantic, which begins to increase at approximately the same time: consistent with previous studies (Zhu and Liu, 2020) which find a 'pile up' of salinity in the South Atlantic as a result of AMOC slowdown. The AMOC in our simulations is too weak, with a preindustrial mean of approximately 7.5Sv at 26°N, and a maximum value of 13Sv in our COU run, declining to approximately 4.5Sv by 2099, as compared to approximately 15Sv in HadGEM2-ES (Martin et al., 2011) and 18±4.9Sv observationally (Johns et al., 2011). This AMOC strength at 26°N in HadGEM2-ES itself is towards the weaker end of estimates from CMIP5 models (Weaver et al., 2012). However, the heat transport is realistic, with a control run heat transport of 0.075PW/Sv at 26°N, as compared to observations of 0.079PW/Sv (Johns et al., 2011). The decline in AMOC strength in our ocean only simulations and HadGEM2-ES simulations are also proportional: over an RCP8.5 scenario, Sgubin et al. (2014) found a decline of AMOC strength at 26∘N from approximately 15.5 to 8Sv at 26°N in HadGEM2-ES.

To explicitly test whether the redistribution of heat from the North Atlantic, and salinity to the South Atlantic, can be explained in terms of a changing AMOC, we calculate the redistribution of heat and salinity through the Equator in the Atlantic. This is calculated as the difference in meridional velocities between the COU and CTR runs, multiplied by the control run temperature and salinity fields (this analysis is conceptually similar to that performed by Williams et al. (2021) in order to calculate the redistribution into/out of a volume, though here we consider only the equatorial boundary between the North and South Atlantic). For the period 1950-2099, for which there are non negligible changes in the redistributed heat content of the North Atlantic, we find the correlation between the redistributed heat content of the North Atlantic and the redistribution of heat through the Equator due to AMOC change has an $R^2$ value of 0.58, suggesting that the changing in overturning circulation plays a key role redistributing heat out of the North Atlantic, and into the South Atlantic. We also find a slightly weaker correlation between the non AMOC driven redistribution of heat past the equator and the North Atlantic heat inventory, with a $R^2$ value of 0.45. These $R^2$ values are reduced to 0.50 and 0.38, respectively, when considering the period 1890-2099.

The picture is similar for salinity: for the period 2000-2099, for which there are non negligible changes in the redistributed salinity content of the South Atlantic, we find a correlation between the redistributed salinity content of the South Atlantic and redistribution of salinity through the equator due to AMOC change has an $R^2$ value of 0.61, which is reduced to 0.04

when considering the period 1890-2099. Changes due to gyre circulation driven redistribution have $R^2$ values of 0.09 and 0.33, respectively, suggesting that the large scale mechanisms of salinity redistribution differ from those of heat.

As this method of calculation is able to infer redistribution directly from model outputs, we have good confidence our decomposition is reliably identifying excess heat and salinity. We therefore believe the redistribution of heat out of the North Atlantic and salinity into the South Atlantic are driven predominantly by AMOC variability, with non AMOC circulation changes influencing the redistribution of temperature and salinity differently. Identifying whether the lack of correlation between our estimates and the explicitly calculated redistribution when there is no appreciable accumulation of either is due to

inaccuracies in our approach or the dominance of other factors in the redistribution of heat and salinity would likely improve our understanding of the strengths and weaknesses of this method, but is beyond the scope of this study.

As with the global inventories, we find little evidence of 'settling' into a new circulation state: in most basins, redistributed heat and salinity inventories continue to grow during our simulations, and AMOC strength declines continually throughout the 21$^{st}$ century. A notable exception is the South Pacific, for which the redistributed heat inventory increases to approximately

450     -50ZJ by 2050, before remaining at a similar value for the rest of the simulation.

One way of assessing the interaction of excess and redistributed heat is to plot changes in their accumulation against each other, with emergent relationships consistent with coupling between the two: this is shown in Figure 7.

In the North Atlantic, we find an acceleration of the accumulation of redistributed heat with respect to the excess heat inventory (Figure 7g). However, in all other basins for which relationships emerge clearly, the accumulation of excess and

redistributed heat are either linearly related (Subpolar Southern (7a), North Pacific (7e)), or sublinear. This is as expected: the acceleration of the accumulation of redistributed heat is unique to the North Atlantic. In all basins other than the North Atlantic, the rate of accumulation of redistributed heat with respect to excess heat slows over the timeseries.

Despite this slowing, the redistributed heat inventories continue to grow, except in the Subantarctic Southern, South Atlantic, South Pacific and Arctic. In all other basins, we see the continued accumulation of redistributed temperature, indicating the

continual dynamic readjustment of the ocean, at an inter-basin scale: the lack of growth at a basin scale imposes no constraints on intra-basin redistribution. Of these, the Subpolar Southern and North Atlantic are the most striking, with heat redistribution increasing linearly and with the square of excess heat accumulation, respectively.

Previous studies have found AMOC strength to be proportional to SST anomalies in the North Atlantic (Caesar et al., 2018), and SST anomalies are thought to be proportional to excess heat (MacDougall and Friedlingstein, 2015). Though it would

initially appear that this would act to linearly couple the excess heat content of the North Atlantic to the redistribution of heat out of the North Atlantic, the redistributed heat inventory will be proportional to the time integrated changes in AMOC strength. The excess heat inventory of the North Atlantic increases monotonically with time, and so the rate of change of the redistributed heat inventory will be proportional to the excess heat inventory. The proportionality of the redistributed heat inventory of the North Atlantic to the excess heat inventory can therefore be explained in terms of the unique circulation of the

North Atlantic.

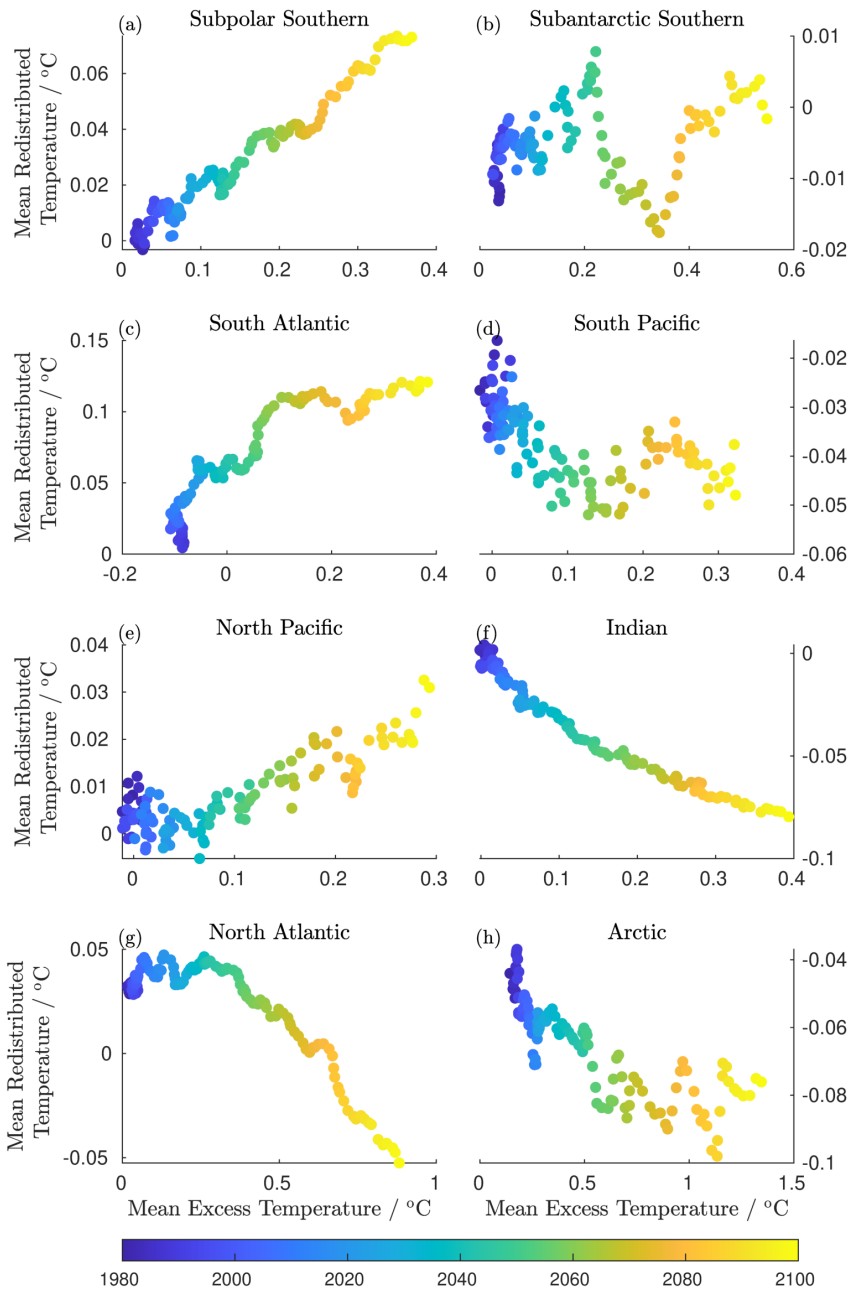

**Figure 7.** The emergent relationships (if any) observed between excess and redistributed heat in each of the 8 ocean basins shown in Figure 7, presented in terms of the mean redistributed and excess temperature changes for the basin. Timeseries begin in 1980 as there is no appreciable accumulation of excess or redistributed heat in the first half of the run. Scales differ for each basin.

### 3.3 Mapping storage of excess and redistributed temperature and salinity

The regional patterns of decadal mean excess and redistributed temperature for the 2090s at the surface and at 2000m is shown in Figure 8 and the regional patterns of the 2090s decadal mean excess and redistributed salinity in Figure 9. For both temperature and salinity, surface changes are dominated in most locations by the excess component. Excess temperatures are positive nearly everywhere, whilst excess salinity is generally positive in the South Atlantic, Subtropical North Atlantic and Indian Oceans, with the Pacific generally negative. This is consistent with increased evaporation over the Atlantic and increased atmospheric freshwater transport from the Atlantic to the Pacific.

It is generally expected that in a warming climate, the hydrological cycle will become amplified, with increased evaporation (precipitation) in regions of net evaporation (precipitation) (Durack and Wijffels, 2010), (Zika et al., 2018), (Gould and Cunningham, 2021). Thus, salty regions of the ocean surface become saltier, and fresh regions fresher. As these changes result from changing surface fluxes, hydrological amplification should be captured by the excess salinity at the surface, rather than redistributed salinity: this is consistent with our results.

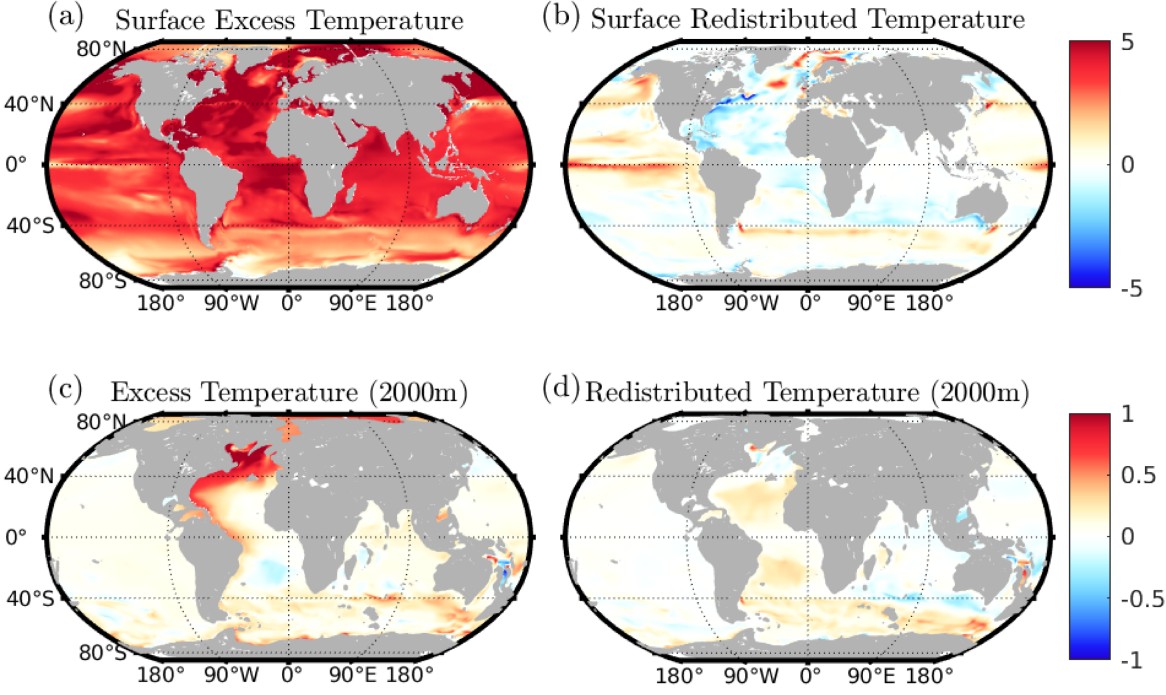

**Figure 8.** Maps of excess and redistributed temperature on two depth surfaces: the surface and at 2000m. Values given are the decadal mean for the decade 2090-2099. Colour axes are shared between each component at both depths.

Whilst surface warming is unsurprisingly dominated by excess temperature, at 2000m the contributions of excess and redistributed temperature to total temperature change are of comparable magnitude, with the exception of the North Atlantic. In

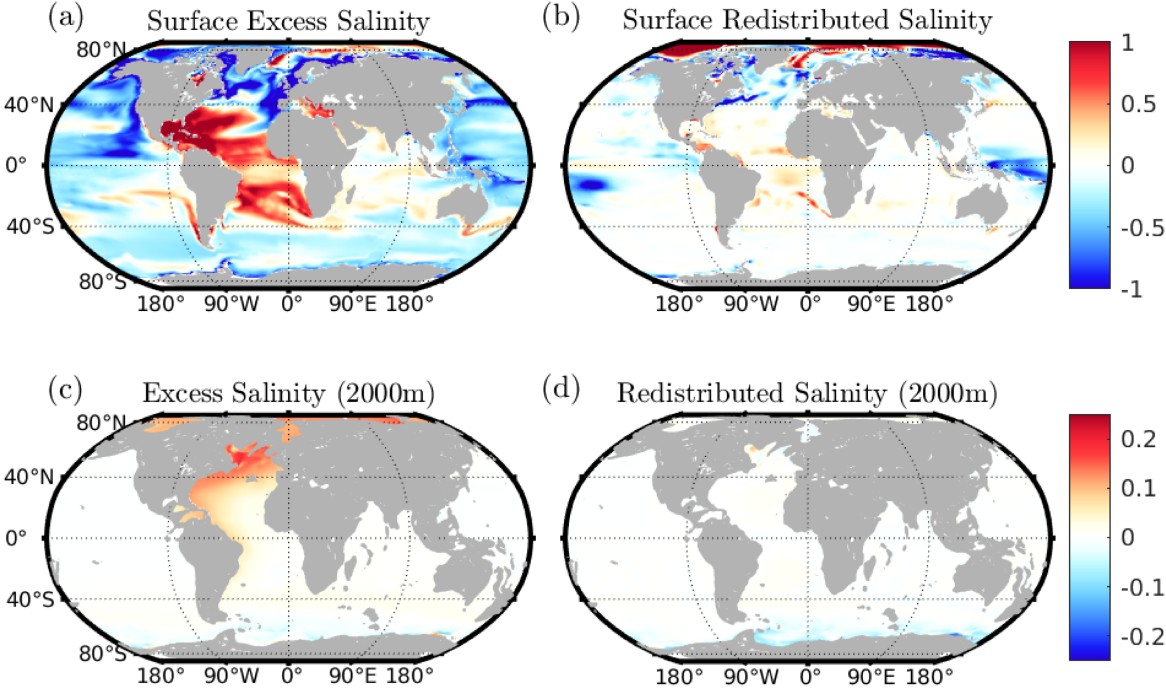

**Figure 9.** Maps of excess and redistributed salinity on two depth surfaces: the surface and at 2000m. Values given are the decadal mean for the decade 2090-2099. Colour axes are shared between each component at both depths.

contrast, the majority of salinity change at depth is accounted for by the excess component, though appreciable changes are generally only found in the North Atlantic. This salinity increase at depth is despite surface freshening in the Subpolar North Atlantic (Figure 9a, 9c), resulting from the propagation of surface salinification here in the 20[th] century. Patterns of excess and redistributed surface salinity are consistent with the results of Sathyanarayanan et al. (2021) and Levang and Schmitt (2015).

The strong surface redistributed salinity signal in the Arctic appears to result from reduced sea ice freshwater transport from the marginal seas of the Arctic inwards. Previous studies using the NEMO GCM coupled to the LIM2 sea ice model have found that Arctic sea ice tends to grow along the coastal shelves of the Arctic Ocean, before being transported by the Beaufort Gyre circulation and transpolar drift (Moreau et al., 2016). The net result of this is to transport both freshwater and DIC from the coastal shelves to the centre of the Arctic Ocean: changes in this transport will therefore act to cause large and tightly correlated changes in DIC and salinity in the surface Arctic Ocean. Our decomposition therefore partitions salinity change resulting from changes in this transport to redistribution. Similar changes in sea ice transport also act to cause redistributed freshening in the coastal Southern Ocean.

The total inventory change in heat, salt, and density by the last decade of our simulation, as well as the storage of the excess and redistributed components are shown in Figure 10, for the upper 2000m of the ocean. We present these as contributions

to steric sea level change, allowing for both normalisation and a comparison of contributions to steric sea level rise. On this timescale, excess (Figure 10c) and total (Figure 10a) heat inventory changes are positive nearly everywhere, with the exception of the Weddell and Ross Gyres. Redistributed heat inventories are negative generally in the North Atlantic (Figure 10c), with the largest values seen in the Labrador and Norwegian Seas, as well as the Subtropical Gyre. In the Pacific and Indian Oceans, redistributed heat inventories are most negative at around 30-35°S.

Salinity inventory changes show a different geographical distribution: excess salinity increases uniformly only in the Atlantic and Arctic oceans (Figure 10e). Total salinity change is again dominated by the excess here. As with heat, the fingerprint of AMOC slowdown can be seen in the redistributed salinity signal: we observe redistribution driven cooling and freshening in the North Atlantic and redistribution driven warming and salinification in the Equatorial and South Atlantic, resulting from a weakening in the northward transport of heat and southward transport of fresh water. This redistribution driven cooling and freshening acts to oppose the warming and salinification associated with increased surface heating and concurrent increases in evaporation - precipitation (E-P).

Density inventory changes (Figure 10a) are relatively globally uniform compared to the individual contributions: a decrease is seen in the total change and excess inventory nearly everywhere, with the exception of the Weddell and Ross Seas, as well as the central Arctic Ocean. The Arctic Ocean decrease is dominated by the changes in freshwater transport, whereas the Weddell and Ross Sea decrease result from upwelling cool water. In the Atlantic, large changes in steric sea level resulting from excess temperature are significantly reduced by the accumulation of excess salinity, and a similar cancellation is seen in the redistributed components.

## 4   Discussion and Conclusions

We have demonstrated a new technique for estimating the redistribution of heat and salinity by the ocean in response to anthropogenic climate change, allowing us to identify the excess signal and producing estimates consistent with other reconstructions. This method can be thought of as sitting within a family of techniques which aim to understand ocean circulation changes through the relationship between ocean temperature and DIC, along with the methods of Bronselaer and Zanna (2020) and Williams et al. (2021). It produces results which are consistent with the assumptions of both methods, without constraints to enforce this. Instead, we assume that on decadal and subdecadal timescales, local ocean heat and carbon content are dominated by redistribution, and that on longer (multidecadal to centennial) timescales, circulation variability dominates over biological changes in natural carbon. This first assumption is consistent with the results of Thomas et al. (2018), who investigated the relationship between ocean heat and carbon content, finding the two to be anticorrelated on decadal timescales. The results of Williams et al. (2021) suggest the assumption of circulation variability dominating over biologically driven changes is also reasonable. A key strength of this new technique is that it also allows us to estimate not only the redistribution of heat, but also salinity, and we see no theoretical reason why it may not be extended to other tracers whose distributions evolve in response to anthropogenic climate change. Furthermore, its implementation is such that in order to identify circulation driven changes in a given tracer requires only timeseries of the tracer in question, and a tracer which we may assume to change distribution

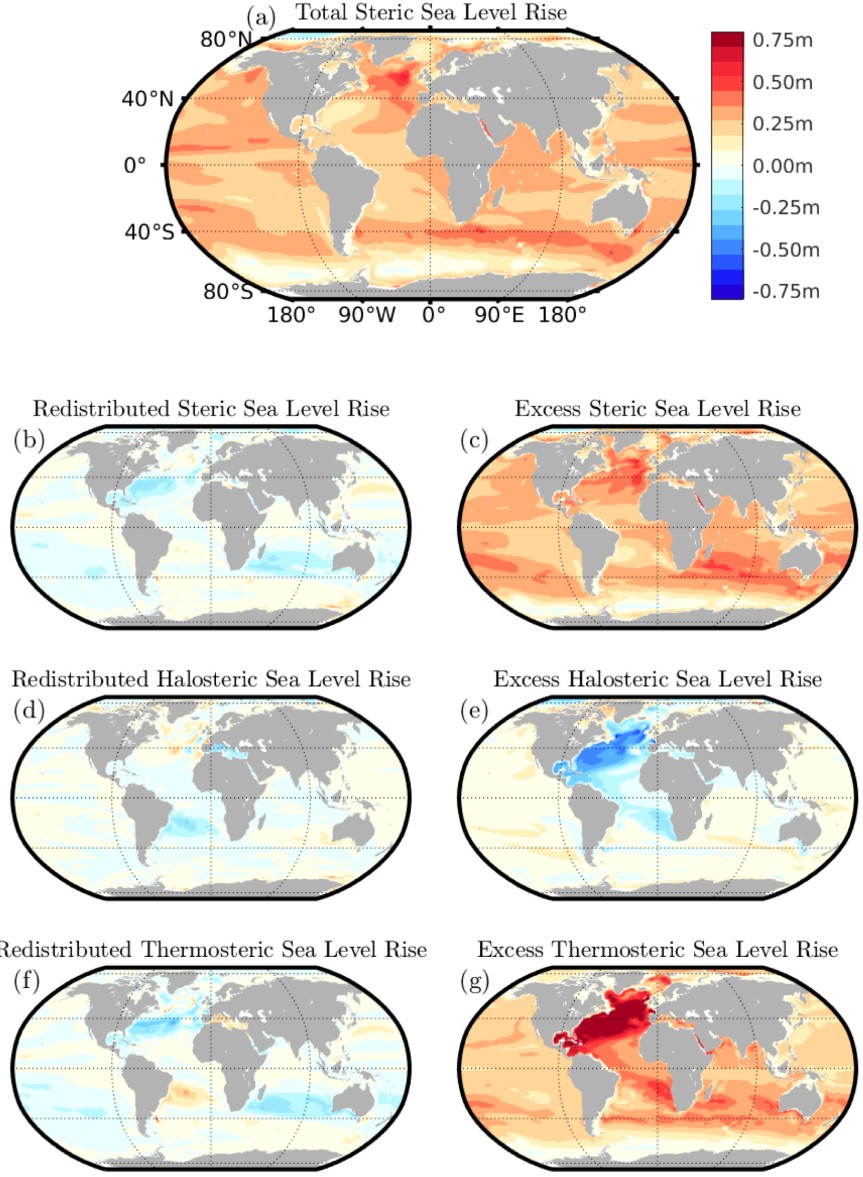

**Figure 10.** 2090's mean steric, halosteric and thermosteric contributions to sea level rise, as well as the total, from the upper 2000m of the ocean.

only through redistribution, for example $C_{nat}$. It should therefore also be applicable to observational timeseries with little modification.

Our globally integrated estimates indicate that magnitude of the excess and redistributed temperature signals are currently of a similar size, with the magnitude of excess temperature signals expected to exceed those of redistributed temperature signals towards the end of the 2020's. This is in keeping with previous studies which find excess heat beginning to dominate over redistributed heat in the period 2011-2060 (Bronselaer and Zanna, 2020). Of course, as this is only one climate change run from a single model, there is a large uncertainty associated with this and we recognise that it does not account for the spread of model responses to imposed climate change under an RCP8.5 scenario. However, our results are internally consistent, demonstrating a number of phenomena thought to occur under a changing climate explicitly in terms of the accumulation of excess heat and redistribution of preindustrial heat.

We have also produced, to our knowledge, the first modelled estimates of the redistribution of the preindustrial salinity field by the ocean and so the excess salinity field: that is, the changes in salinity due to changes in the balance of surface freshwater transport, directly excluding changes in ocean freshwater transport. By extension, we have also been able to produce estimates of excess and redistributed density, and so the contributions to steric sea level rise of temperature and salinity changes. We find that the penetration to depth of the redistributed salinity signal is far weaker than that of temperature, which, with the exception of the North Atlantic, accounts for a similar fraction of deep temperature change as the excess. However, we do find several signals in surface excess and redistributed salinity changes consistent with hydrological amplification, as well as a salinity signal in the South Atlantic as a previously identified 'salinity pile up' in the South Atlantic consistent with AMOC slowdown (Zhu and Liu, 2020). By the 2090's, the Southern and Subtropical North Atlantic show increasing redistributed surface salinity as a result of AMOC slowdown, with a decreasing redistributed salinity in the Subpolar North Atlantic. At the surface, we find that the majority of salinity change results from changes in E-P (excess), rather than circulation changes (redistributed), and that these patterns in excess salinity are consistent with both historical observations globally (Durack and Wijffels, 2010), and, in the Atlantic, with the salinity response to an idealised surface heat flux (Zika et al., 2018). We find that the decrease in global mean excess salinity occurs earlier than the increase in globally integrated excess heat, consistent with previous studies which find significant sea ice loss even in the early 20[th] century, before appreciable global warming (Wadhams and Munk, 2004), (Hetzinger et al., 2019). These results suggest that historical observations of temperature changes are dominated by redistribution, with excess temperature likely to dominate in the coming decades. Historical changes in salinity however may instead be predominately the result of excess salinity, rather than redistribution. This holds at both global and local scales, with the patterns of local excess salinity appearing to be dominated by amplification of the hydrological cycle, and is in agreement with the findings of Stott et al. (2008), Terray et al. (2012), Pierce et al. (2012) and Skliris et al. (2014), who suggest the salinification of the subtropical North Atlantic and freshening of the Western Pacific Warm Pool may constitute an early fingerprint of anthropogenic forcings.

In applying our technique to the Atlantic, we have shown explicitly the redistribution of heat associated with changes to the overturning circulation, in addition to the aforementioned salinity signal. We also find fingerprints of AMOC change in both the redistributed temperature and salinity inventories of the North and South Atlantic: a large and rapid accumulation of negative redistributed heat in the North Atlantic over the period 2037-2099, as well as the accumulation of a large inventory of redistributed salinity in the South Atlantic over the same period. Over the period 2023-2099, for which the accumulation

of excess heat in the North Atlantic is distinct from noise, we find $25\pm2\%$ of global excess heat accumulation is in the North Atlantic. This is remarkably similar to observational estimates of anthropogenic carbon uptake (Sabine et al., 2004), again indicating the close relationship between excess heat and anthropogenic carbon.

By the end of the 21$^{st}$ century heat storage is dominated by excess heat. In addition, excess salinity storage is also largely spatially uniform, though the contributions of redistributed and excess salinity to halosteric SLR are of similar scales in most locations (Figure 10). The only exception to this is a large increase in excess salinity in the Atlantic, where excess salinity inventories are much larger than redistributed salinity inventories. The similar contributions of excess and redistributed salinity storage is despite patterns of regional change in sea surface salinity and salinity inventory changes being dominated by the excess component, both historically and by the end of the 21$^{st}$ century.

By combining our estimates of excess temperature and salinity, we can directly compute the excess density change, and the redistribution of density. In the North Atlantic, we find warming and salinification in the excess components, and cooling and freshening in the redistributed components. In both cases, these changes are in a density compensating fashion. Previous studies have noted that whilst density compensated water mass changes may be a general property of the ocean, the behaviour is particularly marked in the Atlantic (Lowe and Gregory, 2006), as well as important for contemporary Atlantic deep ocean heat uptake (Mauritzen et al., 2012), though it is uncertain how this will evolve. Our results suggest that in the Atlantic, even by the last decade of our simulations, changes in excess temperature and salinity act in a density compensating fashion. A consequence of this is that changes in surface freshwater fluxes associated with climate change oppose the reduction of overturning circulation associated with increased surface warming, opposing the reduction in the North Atlantic's capacity to sequester excess heat. This suggests that the excess contribution to themosteric SLR in the Atlantic will continue to grow on centennial timescales, assuming continued $CO_2$ emissions, though the thermosteric SLR is greatly ameliorated by halosteric sea level fall. This is in agreement with historical observations (Antonov et al., 2002). However, the much smaller redistribution contribution to density indicates that changes to ocean circulation will have little effect on steric SLR in the North Atlantic by the end of the 21$^{st}$ century, although redistributed density compensation in the North Atlantic begins to break down in approximately 2050, as the redistribution of heat out of the North Atlantic significantly exceeds that of salinity by this time.

Finally, although only being applied within a single model, our patterns of excess and redistributed heat storage are consistent with previous studies (Winton et al., 2013), (Bronselaer and Zanna, 2020), (Williams et al., 2021), despite differing assumptions used in the calculation of the redistribution of heat from carbon. A key benefit of the method introduced here compared to prior carbon based estimates of circulation change is that it is potentially applicable both to observations and multiple tracers. In combination with other techniques, we believe this method to be a powerful tool for understanding causes of future ocean temperature, salinity and density change.

*Code and data availability.* The full model outputs used to decompose the temperature and salinity fields are available upon request. Core functionality which allows users to reproduce the decomposition is freely available at https://github.com/charles-turner-1/temp_decomp. This GitHub repository also contains sample cases demonstrating the decomposition, as well as the MATLAB code used to produce the

decomposed fields used in this study: however, reproducing the full fields will require the full model outputs temperature, salinity & DIC fields. This code was run using MATLAB R2020b on Manjaro Linux: modifications to the code may be necessary on other operating systems or using other MATLAB versions.

## Appendix A: Uncertainty in estimates of local redistribution

We estimate a local gradient, $\Delta\theta/\Delta C_{\mathrm{nat}}$ or $\Delta S/\Delta C_{\mathrm{nat}}$ by applying two dimensional PCA to the timeseries of yearly deviations of the two variables from their decadal mean values at each grid cell. This is equivalent to performing a total least squares fit to obtain a linear relationship between the two variables.

We then scale the data to normalise the ranges of $\theta/S$ and $C_{\mathrm{nat}}$ before again performing 2D PCA on our timeseries at each grid cell to estimate the fraction of the covariance contained within each principle component. This yields the fraction of the total variance explained by each principal component, which we refer to as $\varepsilon_1$ and $\varepsilon_2$: these can be thought of the axes of an ellipse describing a scatter cloud relating the two variables. A fit which is a perfect line can be thought of as the limit of this ellipse where $\varepsilon_1 \to 1$ and $\varepsilon_2 \to 0$. Conversely, an essentially random fit through a spherical cloud of points can be thought of as the case where $\varepsilon_1 = \varepsilon_2$.

We use the eccentricity of this ellipse as a suppression factor, $\phi_u$:

$$\phi_u = \sqrt{1 - \left(\frac{\varepsilon_2}{\varepsilon_1}\right)^2} \tag{A1}$$

The need for conservative estimates of confidence in the fit is particularly important for fits in which no discernable correlation can be drawn: for these, gradients associating minor changes in $C_{\mathrm{nat}}$ with large changes in $\theta$ or $S$ can be obtained, effectively at random, and so our suppression factor must remove these effectively. As we concern ourselves primarily with inventories, this approach was found to be preferable to including large uncertainties due to a small number of spurious points, or simply setting a threshold below which we do not attempt to diagnose the redistribution of heat. Only 6% of $\varepsilon_1$ values are scaled by a factor of $1/2$ or less: this was found to be a suitable compromise, with only the most unreliable estimates strongly suppressed.

Alternative methods may produce better quantifications of uncertainty, though are not considered here as the eccentricity method was sufficient for our purposes.

We then calculate the redistribution coefficient $\kappa_r$ as

$$\kappa_r = \phi_u \times \kappa_s \tag{A2}$$

The implementation of this is demonstrated in Figure A1, for two points in the North Atlantic at approximately 24°N, 30°W and 850m and 1950m. The poorly correlated point, Figure A1a and A1c, is an extreme outlier, shown for demonstrative purposes. Here the fit is essentially random, and so estimates of temperature redistribution are scaled to reflect this uncertainty: the eccentricity of the ellipse described by the cloud of points in $\theta - C_{\mathrm{nat}}$ space is used as a scale factor. For the strongly correlated point, shown in panels (b) and (d), temperature and $C_{\mathrm{nat}}$ variability are almost perfectly anticorrelated, representing the dominance of vertical structure in determining the redistribution coefficient $\kappa_r^T$. Here, $\kappa_r^T = -0.0210$, $\partial_z\theta/\partial_z\mathrm{DIC} = -0.0208$.

## Appendix B: Merging one and two step estimates

Our estimation technique assumes that the relationship between short timescale changes is dominated by circulation variability.

However, at the surface, changes in salinity and $C_{nat}$ are instead dominated by freshwater fluxes: an excess of evaporation over precipitation will increase concentrations of salt and $C_{nat}$, coupling changes in the two. This leads to changes which are properly described as excess salinity being partitioned into redistributed salinity.

To account for this, we use a two step estimation process. As we note, we may combine Equations 8 & 9 in order to estimate redistributed salinity from redistributed temperature, or vice versa. We therefore estimate excess salinity at the surface as

$$\Delta S_r = \kappa_r^{\text{T}-\text{S}} \times \Delta\theta_r, \tag{B1}$$

where $\kappa_r^{\text{T}-\text{S}}$ is an estimate of the local slope of the temperature-salinity curve, produced in the same fashion as our previous estimates. We refer to this estimate of surface excess salinity as a two step estimate. ? We then merge the two estimates using a sigmoidal weighting scheme based on depth. Our simulations use 64 vertical levels, with the 20th level corresponding to approximately 200m. Denoting the $i^{\text{th}}$ vertical level $z_i$, the one step estimate as $S_1$ and the two step estimate as $S_2$, we

calculate our final estimate of salinity redistribution, $S$, as

$$S = S_1 \times \sigma\Big(\frac{z_i + 20}{2}\Big) + S_2 \times \Big(1 - \sigma\Big(\frac{z_i + 20}{2}\Big)\Big), \tag{B2}$$

where $\sigma(z)$ is the sigmoid function:

$$\sigma(z) = \frac{1}{1 + e^{-z}} \tag{B3}$$

## Appendix C: Gamma Factor

Figure C1 shows the $\gamma$ factor over our full run. It increases from 0 at the beginning of the run to 0.117 by 2099. We perform smoothing as in the late $19^{\text{th}}$ and early $20^{\text{th}}$ century, $C_{anth}$ inventories are small and so large corrections are necessary to perfectly correct a small amount of $C_{sat}$ outgassing: smoothing removes this effectively. By the $21^{\text{st}}$ century, $C_{anth}$ inventories are large enough that smoothing has little effect. Finally, we note that the $\gamma$ factor does not begin to increase significantly until the late $20^{\text{th}}$ century, approximately the same time that globally integrated ocean heat content begins to increase. Thus, to first

order, $\gamma$ corrects for $C_{sat}$ outgassing due to ocean warming.

## Appendix D: Patterns of $\kappa_r^T$, $\kappa_r^S$

Figure D1 shows patterns of $\kappa_r^T$ for the Atlantic (a), Indian (b), and Pacific (c) oceans, and Figure D2 the patterns of effective $\kappa_r^S$ for these basins. Patterns of $\kappa_r^T$ are calculated directly. As our redistributed salinity estimates are produced from the combination of a one step and two step estimate (as described in Appendix B), the patterns of $\kappa_r^S$ are instead a map of ef-

fective values. These are calculated by diagnosing the mean redistributed salinity for the decade 2090-2099 (this was chosen

to maximise changes to adjusted $C_{nat}$ and thus avoid numerical issues), before dividing by mean changes in adjusted $C_{nat}$ to obtain a map of effective $\kappa_r^S$ values. Thus, these values are identical (to numerical precision) to those calculated directly in the mid-depth and deep ocean, but represent a best estimate of the spatial coupling between salinity and natural carbon in the upper 200m of the ocean, and avoid the complicating effects of freshwater dilution.

## Appendix E: Redistributed heat content of the unventilated ocean

In the absence of simulations explicitly including PAT, we investigate the accuracy of the decomposition and the appropriateness of Equation 8 by examining the total and redistributed heat content change of a region of ocean for which ventilation is negligible; here, excess heat content is assumed to be zero and heat content change and redistributed heat content change are expected be identical. Thus, the accuracy of the decomposition can be examined, without the need for simulations including PAT.

We identify the unventilated region of the ocean as all grid cells in our model runs for which $C_{anth}$ concentrations do not exceed a given threshold at any time during the model run. We use a cutoff $C_{anth}$ concentration of $|1\mu mol/kg$ to define the unventilated ocean. In addition, we include grid cells only in the depth range 2000-3000m in this region: this excludes grid cells for which large spurious, negative $C_{anth}$ values are seen (See Figure 2).

Figure E1 shows the total heat inventory change, and our reconstruction redistributed heat content for this region of unventilated ocean: for a perfect reconstruction, we expect these quantities to be nearly identical, and this is indeed the case, in particular for higher frequency variability (E1a), and for longer term variability before approximately 2050 (E1b). However, over the full model run, our estimate begins to systematically diverge, with our estimate of mean temperature change due to redistribution slightly lower than the mean temperature change.

To determine the quality of our reconstruction, we compute Taylor Skill Scores (Taylor, 2001) for the periods 1890-1950, and 1890-2099. Following Hirota et al. (2011), these are calculated as

$$S = \frac{(1+R)^4}{4(\hat{\sigma_f} + 1/\hat{\sigma_f})}, \tag{E1}$$

where $S$ is the Taylor Skill Score, $R$ the Pearson correlation coefficient, and $\hat{\sigma_f}$ the ratio of the model output timeseries standard deviation ($\sigma_f$) and the reconstruction timeseries standard deviation ($\sigma_r$):

$$\hat{\sigma_f} = \frac{\sigma_f}{\sigma_r}. \tag{E2}$$

For the period 1890-1950 (Figure E1a), before meaningful climate change, we obtain a Taylor Skill Score of 0.838: thus representing skillful reconstruction of the redistributed heat content.

Over the full time period (Figure E1b), our reconstruction again captures the higher frequency variability in heat content of the unventilated region well, though the rate of accumulation is somewhat lower: the ratio of mean temperature change to mean redistributed temperature asymptotes to approximately 80% over the full model run. Over this period, we obtain a slightly higher Taylor Skill Score of 0.964, indicating our reconstruction is capturing the redistribution of heat into this region

accurately on longer timescales, although again biased low as with the shorter timescale changes. The decomposition is thus thought to capture approximately 80% of the forced circulation change.

We note however that our definition of unventilated waters (Canth < 1 $\mu$mol/kg) may be too expansive, and thus suggest that 20% uncertainty detailed here represents an upper estimate of the error introduced by our method: as our cutoff is nonzero, some excess heat will enter this region, which will act to systematically increase the total heat content in this region beyond the redistributed heat content.

Finally, we also note that over the period 1890-2050, we obtain a Taylor Skill Score of 0.987, and that over the period 2000-2050, the mean ratio of total temperature changes to redistribution driven temperature changes is 0.994, suggesting a highly accurate reconstruction.

## Appendix F: Glossary of Terms

For an arbitrary tracer $Q$, transported by a velocity field $\mathbf{v}$, we may write

$$\mathbf{v}Q = (\mathbf{v}_0 + \mathbf{v}')(Q_0 + Q') = \underbrace{\mathbf{v}_0 Q_0}_{Preindustrial} + \underbrace{\mathbf{v}' Q_0}_{Redistributed} + \underbrace{\mathbf{v}_o Q' + \mathbf{v}' Q'}_{Excess}, \tag{F1}$$

where $\mathbf{v}_o$ and $Q_0$ refer to the preindustrial components of the velocity field, $\mathbf{v}$, and the tracer field, $Q$, and $\mathbf{v}'$ and $Q'$ the perturbations. The excess and redistributed changes in $Q$, denoted $Q_e$ and $Q_r$ respectively, are therefore given by

$$Q_e(t) = \int_{t_0}^{t} \left( F_Q' - \left( \mathbf{v}_0 + \mathbf{v}' \right) \cdot \nabla Q' \right) dt, \tag{F2}$$

and

$$Q_r(t) = - \int_{t_0}^{t} \left( \mathbf{v}' \cdot \nabla Q_0 \right) dt, \tag{F3}$$

where $F_Q'$ is the anomalous surface flux in $Q$, $t_0$ is a preindustrial time, and $t$ is a generic time. These definitions are described in further detail in Williams et al. (2021) and below.

- **Excess Q**: Changes in the local ocean $Q$ field value due to the imposition of changes in the surface forcing of the $Q$ field. Excess $Q$ may be positive or negative, depending on changes in surface forcing.

- **Excess Temperature**: Change in local ocean temperature due to changing surface heat fluxes, for example warming due to increased radiative forcing at the sea surface.

- **Excess Salinity**: Change in local ocean salinity due to changing ocean freshwater fluxes, for example salinification as a result of increased evaporation and/or reduced precipitation at the sea surface.

- **Redistributed Q**: Changes in the local ocean $Q$ field value due to changes in ocean transport, either imposed in response to climate change or as the result of natural variability. As redistribution can only rearrange the inventory of $Q$ within

the global ocean, the global ocean inventory of redistributed $Q$ must always sum to zero, as positive redistributed $Q$ in one location must be compensate for by negative redistributed $Q$ in another location.

– **Redistributed Temperature**: Changes in local ocean temperature due to circulation change, for example cooling in the North Atlantic due to the reduction of northward heat transport associated with AMOC decline.

– **Redistributed Salinity**: Changes in local ocean salinity due to circulation change, for example salinification in the South Atlantic due to the reduction of northward freshwater transport associated with AMOC decline.

– **DIC**: Dissolved Inorganic Carbon, also known as $tCO_2$. This is the total local inorganic carbon content. It may be decomposed as

$$DIC = DIC_{sat} + DIC_{carb} + DIC_{soft} + DIC_{diseq} + C_{anth} = C_{nat} + C_{anth} \tag{F4}$$

where

$$C_{nat} = DIC_{sat} + DIC_{carb} + DIC_{soft} + DIC_{diseq} \tag{F5}$$

– **DIC$_{sat}$**: Saturation carbon, the DIC content which a parcel of water would have, if allowed to equilibrate with the preindustrial atmosphere at its potential temperature and salinity. It accounts for the bulk of DIC concentrations, around $2000\mu$mol/kg.

– **DIC$_{carb}$**: Carbonate carbon, DIC content due to the remineralisation of calcium carbonate. Concentrations increase with age, with concentrations up to approximately 60 $\mu$mol/kg in the oldest waters.

– **DIC$_{soft}$**: Soft tissue carbon, DIC content due to the remineralisation of soft tissue. As with $DIC_{carb}$, its' concentration increases with the age of water, up to approximately $200\mu$mol/kg in the oldest waters.

– **DIC$_{diseq}$**: Disequilibrium carbon, the DIC content due to the disequilibrium of a parcel of water with the overlying atmosphere, when subducted away from the surface. It may be either positive or negative.

– **C$_{anth}$**: Anthropogenic carbon, the DIC content of a parcel of water due to equilibration with the increased atmospheric $CO_2$ content of the atmosphere, relative to preindustrial. It is defined as having a preindustrial concentration of zero, and hence is closely related to excess DIC (to see this, let $Q_0 = 0$ in Equation F1.

– **C$_{nat}$**: Natural carbon, the DIC content of a parcel of water with the contribution from increased atmospheric $CO_2$ concentrations removed. It is the sum of the saturation, soft tissue, carbonate and disequilibrium pools of DIC. As excess DIC and $C_{anth}$ are closely related, $C_{nat}$ therefore approximates redistributed DIC. However, in response to a warming ocean, the global $C_{nat}$ inventory will decline, causing it to systematically differ from redistributed DIC.

– **C$_{nat}^{adj}$**: Adjusted Natural Carbon, calculated as $C_{nat}^{adj} = C_{nat} + \gamma C_{anth}$, where $\gamma$ is a factor between 0 and 1. This aims to correct for the outgassing of Saturation Carbon in response to ocean warming, in order to adjust for the systematic

reduction in natural carbon leading to inconsistency in the definition of changes in natural carbon and the redistribution of DIC.

– **Excess DIC**: Changes in local DIC content driven by changes in surface conditions: these include changes to surface wind forcing, SST driven change in $CO_2$ solubility in surface, but predominately those due increases in atmospheric $CO_2$ concentrations.

    – **Redistributed DIC**: Changes in local ocean DIC content due to circulation change, for example an increase in the DIC concentration of the deep North Atlantic due to reduced formation of North Atlantic Deep Water.

*Author contributions.* CT designed the technique for temperature decomposition, and EM identified the extensibility to salinity. All authors contributed to interpretation and writing the manuscript.

*Competing interests.* The authors declare no competing interests

*Acknowledgements.* We would like to thank Matthew Couldrey for kindly allowing us to use his simulations for this research. EM, PB were supported by Natural Environment Research Council grant NE/P019293/1 (TICTOC) CT was supported by the Natural Environmental
Research Council [grant number NE/L002531/1]

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

910

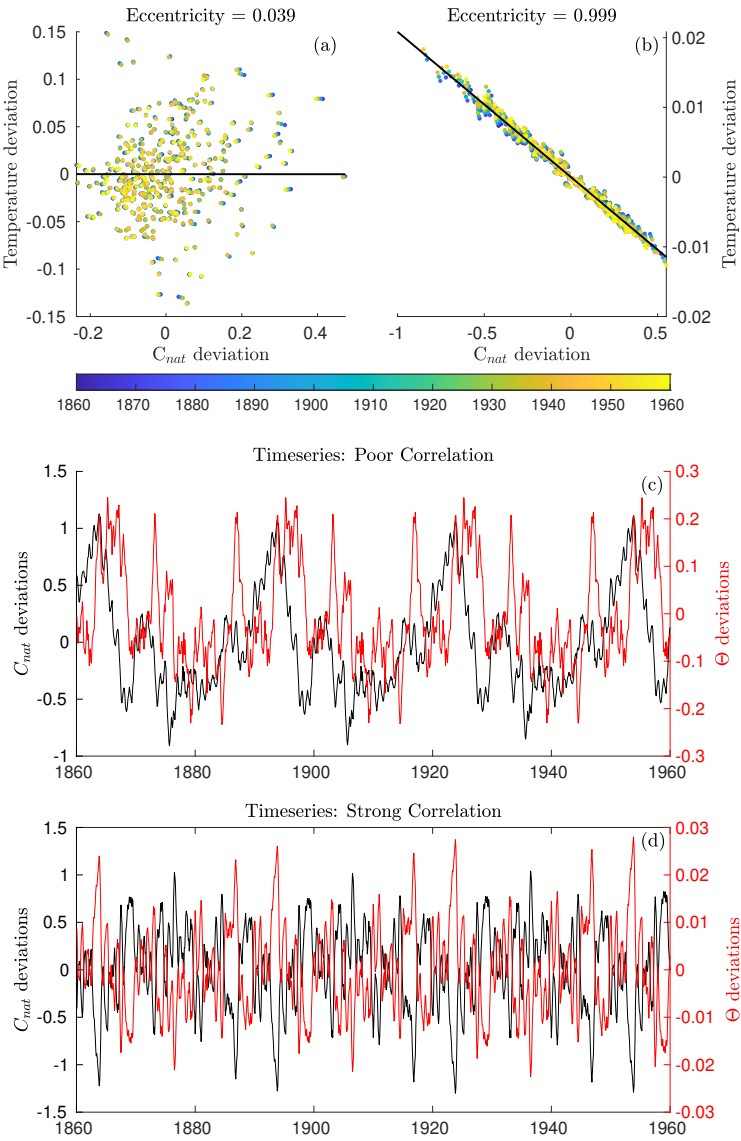

**Figure A1.** The correlations between $C_{nat}$ and $\theta$ used to establish a $\kappa_r$ value, for a poorly correlated point ((a),(c)), and a well correlated point ((b),(d)), in $\theta - C_{nat}$ space ((a), (b)), and timeseries of both ((c),(d)). These two points are located at 24N,30W in the Atlantic, at depths of 850 and 1950m. The major axis of the covariance ellipse in panels (a) and (b) is shown in black.

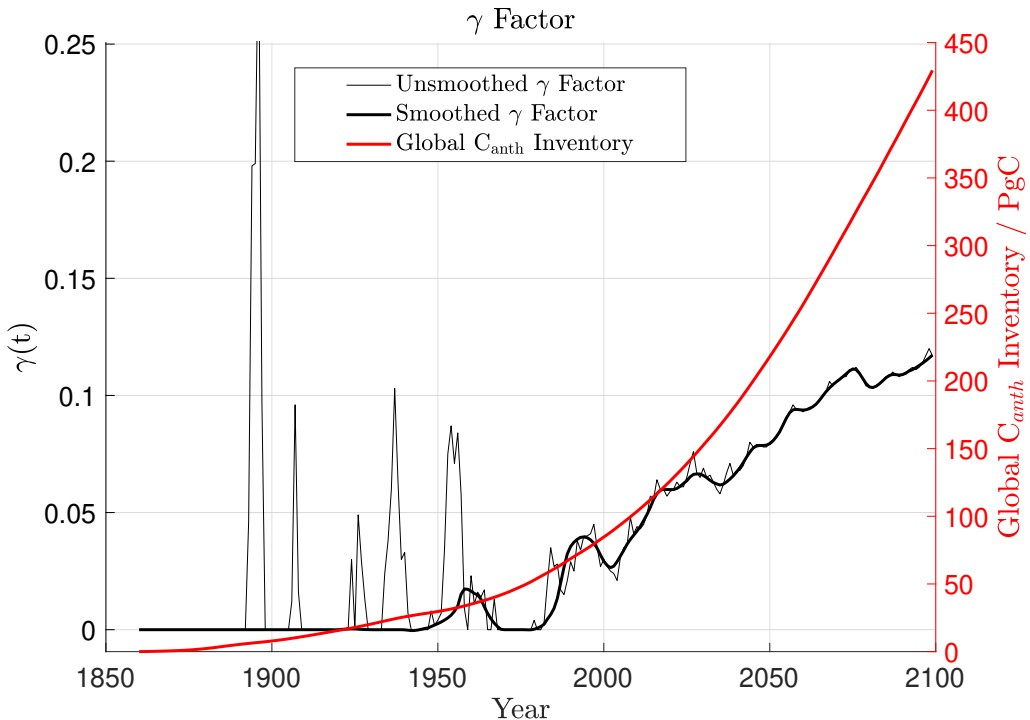

**Figure C1.** The calculated $\gamma$ factor (thin black line), smoothed $\gamma$ factor (thick black line), and global anthropogenic carbon inventory (red line).

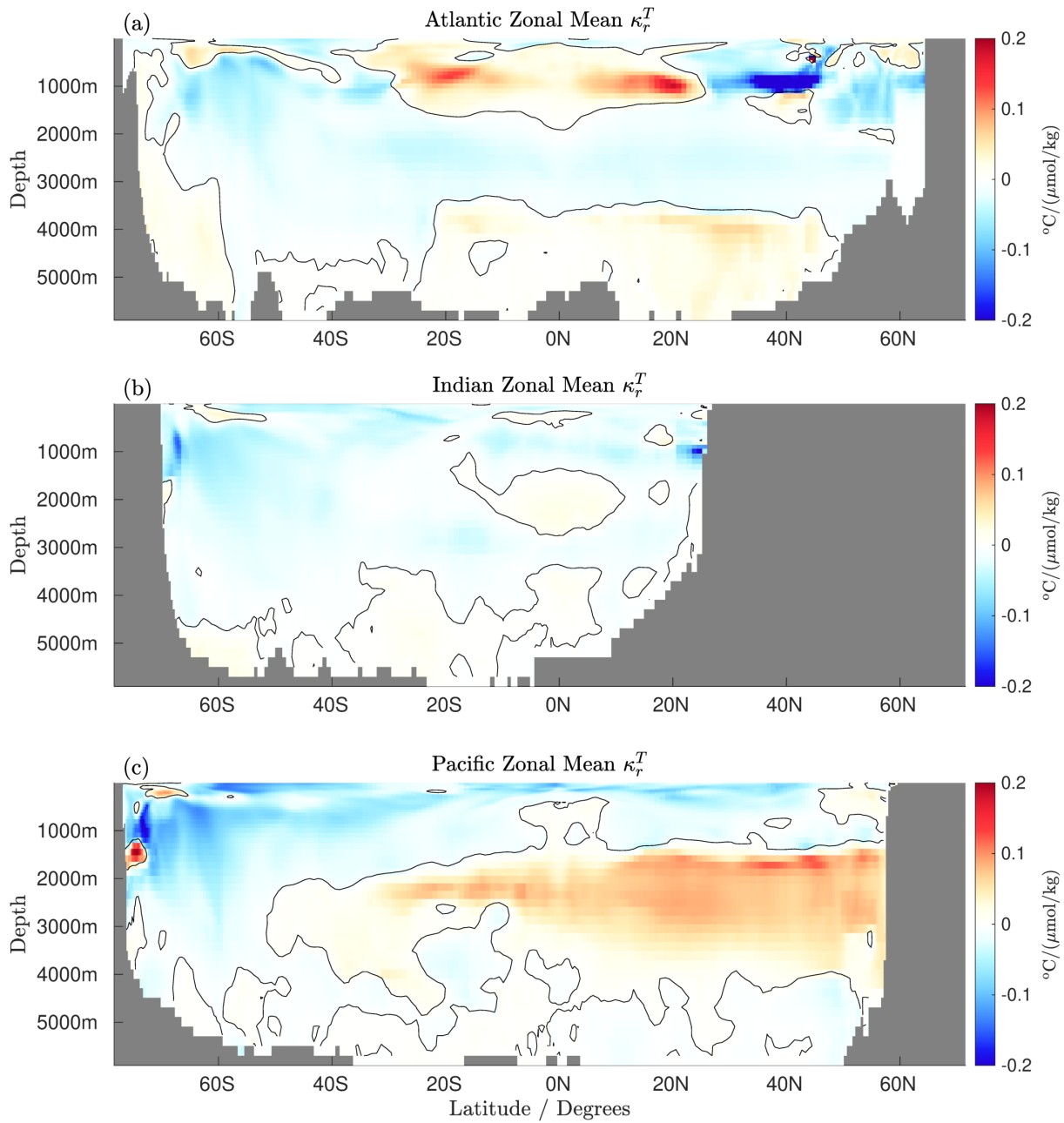

**Figure D1.** Zonal mean $\kappa_r^T$ values for the Atlantic (a), the Indian ocean (b) and the Pacific (c).

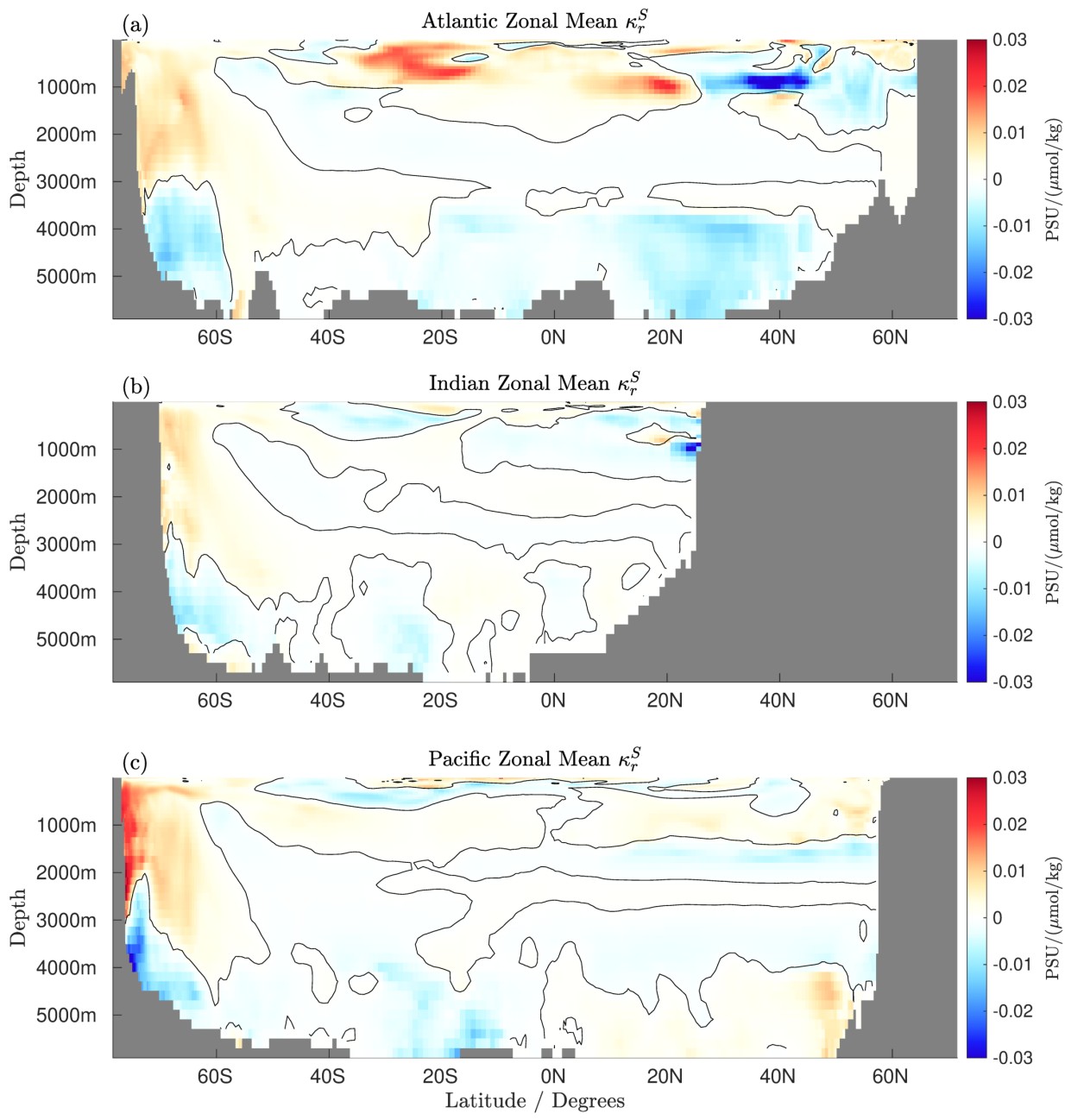

**Figure D2.** Zonal mean effective $\kappa_r^S$ values for the Atlantic (a), the Indian ocean (b) and the Pacific (c).

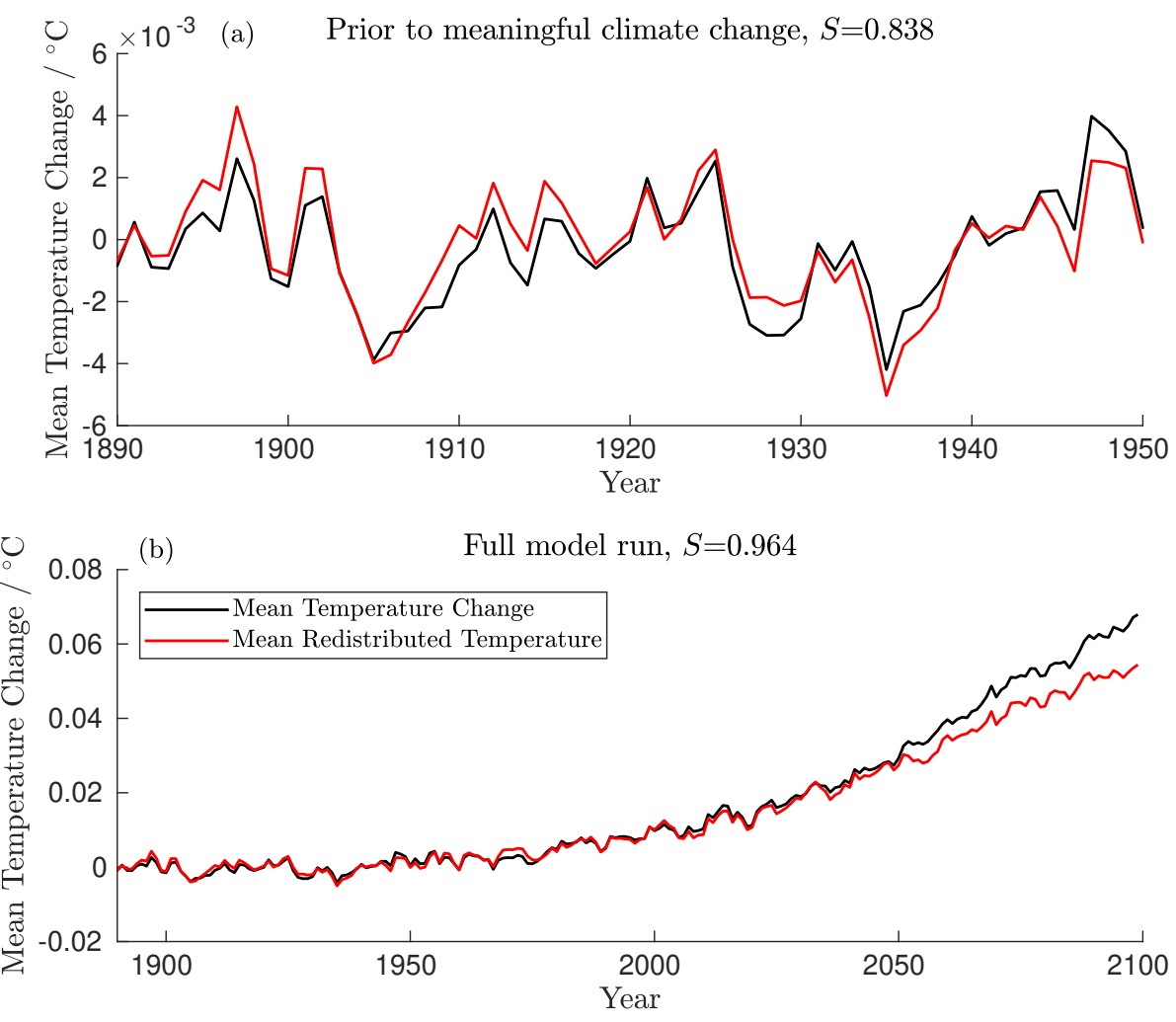

**Figure E1.** Total and redistribution driven changes in the mean temperature of the unventilated ocean for the period before meaningful climate change (a), and the full model run (b). Taylor Skill Scores for the periods presented are shown in subplot titles.