# Peer review of "Decomposing oceanic temperature and salinity change using ocean carbon change"

_Ocean Science, 2021_

## Author Response (AR1)

**1 Reviewer 1**

**1.1 Major Comments**

1. Accuracy. The novelty of the carbon-based methodology described here is that it can be used to derive estimates of excess and redistributed heat (extended to salt) for both model outputs and real-world observations. This latter application will enable ocean temperature time series observations to be deconvolved into that driven by ocean circulation and that caused by uptake of atmospheric anthropogenic heat, something that is simply not possible using PAT. We demonstrate the merits of the approach in a model context as it first allows us to look at global patterns of excess heat, something which is well studied, and also avoids problems associated with observational data sets, namely problems with temporal and spatial resolution, and the lack of a PAT. The reviewer is correct to request better characterisation of the validity of our outputs and given the constraints of our model runs we have given further explanation. Unfortunately due to the setup of our model (simulations were primarily motivated for study mechanisms of carbon uptake) no PAT was included, meaning it is not possible to directly validate our outputs against such an experiment. While this is not optimal we have however now performed a cross validation with the method of Bronselaer Zanna, 2020, who approximated the excess heat field with the anthropogenic carbon field, and were able to validate their approach directly against fixed circulation ocean heat uptake experiments and a PAT (section 3.1, lines 263-343). We show that our results strongly agree, and that the differences between the two methods closely resemble the differences between their method and their fixed circulation heat uptake; we are able to explain this in terms of subduction of carbon through the mixed layer base, indicating differences are likely due to a relaxation of the assumption of a global mean alpha value. This is described in (section 3.1, lines 327-336). As the key motivation of this study is to produce a technique which allows us to estimate excess heat observationally, rather than to model it, this necessitates the use of an alternative methodology to PAT / fixing ocean circulation. We feel we have now better explained how we go about validating its outputs.

2. Consistency. We think the Winton et al. paper cited by the reviewer is in fact Winton et al. 2012: "Connecting Changing Ocean Circulation with Changing Climate", which looked at the patterns of heat and carbon storage under freely evolving and fixed preindustrial ocean circulation experiments. In this paper it is shown that a substantial fraction of redistribution heat transport is compensated for by changes in surface fluxes (around 1/3), indicating that redistribution heat transport is a driver of surface heat uptake from low to high latitude. Following this, the reviewer is correct to question the impact of flux and circulation perturbations on our outputs. Our ocean only simulations have an AMOC strength that is approximately 50% weaker than that in HadGEM2-ES ( 8Sv as compared to 15Sv at 26N in the control simulation; text amended at section 3.2 lines 397-404) to include discussion of this). This would imply that a possible bias in surface heat fluxes associated with redistribution exists due to inconsistency with the forcings. However, we find realistic heat transports by the AMOC in our simulations (when compared to observations - see revised section 3.2 lines 400-402), as well as an AMOC decline in our simulation that is proportional to AMOC decline in HadGEM2-ES, with AMOC strength being approximately half that of HadGEM2-ES at all times in our simulation (section 3.2 lines 402-404). Therefore, although there are likely to be inconsistencies in the forcing and the circulation response, comparisons of the AMOC decline between the ocean-only model and HadGEM2-ES are due to a systematic bias rather than a bias which varies over the course of the runs. Similar such biases resulting from inconsistencies in forcings and sea surface temperatures also exist in FAFMIP type experiments, and are not unique to this study. However, there will still be an excess component of temperature and salinity as a result of surface forcing, and a redistributed component due to changes in circulation; these will be internally consistent within the ocean-only model run, even if the surface forcings are somewhat inconsistent with the atmospheric forcing. In summary, in this study we are looking to describe the utility of using a new method based on carbon data to reconstruct excess and redistributed heat. While the reviewer is correct that the use of an ocean model forced by a coupled model introduces biases, due to the lack of a feedback from the ocean model on the coupled model's atmosphere the existence of these biases does not undermine the demonstration of the method. The text has been amended to account for this discussion (section 3.2 lines 398-400).

3. Representativeness. Models vary in their circulation response to climate change forcings (as well as their uptake of excess heat) that manifest themselves in a range of different temperature changes in various locations in response to increasing radiative forcing. We recognise that we present the outputs of only one model, and we agree with the reviewer that this therefore does not represent the full spread of potential scenarios (for example of AMOC decline) that models predict - we acknowledge this in the revised section 4 lines 516-519, 579-580. However, we don't feel it impacts the utility of the method that we present, or its possible wide application. While we feel applying our decomposition for a range of CMIP5 models is outside of the remit of this particular study, following the reviewer's excellent suggestion we have however subsequently compared the circulation changes in our model with those of HadGEM2-ES, and of HadGEM2-ES with a range of other CMIP5 models, as well as the representation of the AMOC in HadGEM2-ES with observations (section 3.2 lines 397-405).

Minor comments

1. We agree that further clarification is necessary for the lines highlighted. Additional context and explanation has now been added (lines 43-48).

2. A number of techniques for estimating anthropogenic carbon from observations have been derived. These are based either on the inorganic carbon system (C*, TrOCA, phi) or using measurements of transient tracers as proxies of anthropogenic carbon invasion (TTD), and have been used to quantify oceanic anthropogenic carbon accumulation (Khatiwla et al (2013), and its decadal variability (Gruber et al, 2019). As anthropogenic carbon is closely analogous to excess carbon (Williams et al 2021), we thus have better estimates of excess carbon observationally than excess heat. We have amended the text at lines 59-70 to better explain this.

3. The reviewer makes a good point, we have now significantly expanded the experimental specification in the revised draft, with details of all these included (section 2.1, lines 106-140).

4. Thank you for highlighting this error, this has now been fixed

5. As detailed above when addressing major comment 1 (accuracy) we have now completely rewritten this section (section 2.2, lines 142-260) to make it clearer how our estimation technique works in the revised draft, removing details about how velocity perturbations project into temperature-carbon space, and focussing on how the redistribution of temperature and carbon are linked. We have also cross validated our decomposition against an alternative carbon-based proxy method.

6. We thank the reviewer for this suggestion. We have added observational global excess heat uptake from Zanna et al 2019 to Figure 1b. For excess salinity, the signal is dominated by changes in air-sea freshwater fluxes (evaporation-precipitation) rather than ice melt so comparison to observational ice-melt budgets will not be as useful/powerful as hoped. However, we have adapted the text at lines 351-353 to consider this.

7. The reviewer makes an interesting point. However, although global mean redistributed temperature is constrained to be zero, this is not the case locally or regionally. We therefore think that comparing the scale of positive only and negative only regions is useful, as it is a measure of the degree to which excess and redistributed temperature contribute to local temperature changes.

8. We have adapted the revised draft (lines lines 391-393) to clarify that the AMOC does declines continually throughout the COU run, with the decline not slowing.

9. We have reworded the text to make this aspect clearer (lines 444-451). Rather than proportionality existing between the excess and redistributed pools and SST anomalies, it is the rate of change in the redistributed heat pool that is proportional to SST change/AMOC change.

10. We have removed this discussion from the revised draft, as we have decided that it is too speculative and ambiguous.

11. While our model setup does includes sea ice it does not include ice sheets. This means that unfortunately it is not possible to directly examine the relationship between excess salinity changes and glacial inputs in our model. Following this and as mentioned in point 6 above, the excess salinity signal is dominated by changes in air-sea freshwater fluxes (evaporation-precipitation) so comparison to observational ice-melt budgets will not be as useful/powerful as hoped. References to previous studies finding nontrivial ice sheet loss in the early 20th century was to highlight the different emergence timescales of excess heat and excess salinity. We have amended the text to improve clarity regarding this (lines 371-377, 534-454), and to better describe the model setup (lines 106-140). We have also additionally cited Hetzinger et al. 2019 and Wadhams Munk, 2004 as further support (line 535-537), as Hetzinger et al. reconstructed sea ice decline using an algal proxy and found decline in the Arctic started by the early 1910s. Wadhams Munk calculated a 20th century mean sea ice decline equivalent to a global mean salinity change of $10\hat{}4$ PSU: our simulations give a global mean sea ice decline of 5*10^4 PSU over the same period - of the same order of magnitude.

**2  Reviewer 2**

1. (a) We have now significantly rewritten the text to describe the experimental specification for individual runs in far greater detail (lines 109-124) separating this out into sections for CTR (lines 117-119), COU and RAD (120-124). With regard to the specific questions/points raised by the reviewer: surface heat, momentum, freshwater fluxes and atmospheric chemistry from HadGEM2-ES combined with RCP8.5 atmospheric $CO_2$ concentrations was used to force NEMO, with no restoring used – description of this has been included in lines 111-116. Additional references as suggested by the reviewer to make the experimental specification clearer have been added (line 137). To describe the different runs we follow the terminology employed by Schwinger et al. 2014, namely COU, CTR and RAD. Description of each of these have been added to the text (lines 109-110, 117-122) to aid the reader. For the COU simulation, this specifically refers to a coupled climate change run that imposes both physical and biogeochemical change (line 120-122).

   (b) Following the reviewers recommendations section 2.2 has been completely rewritten to more clearly explain how the decomposition works (lines 142-260) and to show that that it is producing an estimate of how the preindustrial states of temperature and DIC covary. We have removed the confusing formulation about how velocity perturbation project into temperature carbon space and focussed on the key principle of estimating how temperature and DIC covary spatially, adding references as appropriate (lines 142, 144, 200). As detailed in response to Reviewer 1 (Comment 1) the simulations we use here were performed to study mechanisms of carbon fluxes and variability in the carbon cycle. As such, no PAT tracer was included nor fixed circulation experiments were performed, meaning it is not possible to directly validate our outputs against such an experiment. We have now performed a cross validation with the method of Bronselaer Zanna, 2020, who like we said, approximated the excess heat field with the anthropogenic carbon field, and were able to validate their approach directly against fixed circulation ocean heat uptake experiments and a PAT (section 3.1, lines 263-348). We quantify our method against the alternative carbon proxy of Bronselear Zanna 2020 (lines 261-348) and explain the differences between the two methods in terms of oceanographic features. Williams et al. 2021 explained and defined excess and redistributed temperature and DIC in terms of the correlation of the excess and anticorrelation of the redistributed components (lines 86-90). We have amended the text to clarify that our decomposition is a method for specifying the anticorrelation between redistributed components, much as the method of Bronselaer & Zanna 2020 specifies the correlation between the excess components (lines 90-96). In this study natural carbon is defined as that component of the total inorganic carbon signal that would have existed in the absence of an anthopogenically-forced atmospheric $CO_2$ concentration, essentially a preindustrial carbon field (lines 126-136) We define how this is a useful tracer in our technique (lines 95-96, 187-194), qualifying that the choice of natural carbon is not unique (lines 190-194).

   (c) We have clarified that we use PCA in order to obtain a total least squares regression (rather than an ordinary least squares regression) as this allows the relationship defined to be independent

of the choice of dependent variable (lines 211-215). As described at lines 206-209 decadal data binning is performed in order to eliminate the small effects of model drift and using repeat surface forcing cycles that are misattributed as excess temperature/DIC.

(d) We thank the reviewer for this advice. As well as improving the description and application of the decomposition (detailed above, lines 142-260), we have amended the text to clarify that using local relationships between temperature and natural carbon derived from observations we can deconvolve estimates of excess and redistributed temperature (and by extension salinity) (lines 508-512). Carbon is useful observationally as we have global observations of temperature and carbon dating back to 1990 (GLODAP) that enables the development of the excess/redistributed fields to be tracked with time. We have added text to explain that that the choice of carbon is not unique, though carbon is useful because excess/redistributed carbon and anthropogenic/natural carbon are similar, with reference to Williams et al. 2021 (lines 68-70).

2. We thank the reviewer for these constructive suggestions. We have rewritten the methodological derivation section to more effectively describe how and why we define the covariability of the preindustrial states of temperature and carbon (lines 142-260). As detailed in a previous response above, we have removed all descriptions of how velocity perturbations project into temperature-carbon space, as this was an unhelpful formulation which did not make our approach clear. We have removed repetitions, added further references (lines 141, 168, 200). Need a list here of actions taken and line references

(a) We recognise the ambiguity relating to our discussion of salinity and lead/lag times. To account for this we have calculated times of emergence of signals of excess heat and salinity from noise for the basins discussed (lines 371-377) which are presented in Figure 6. We have also clarified the text to reflect that the discussion primarily concerns the emergence of signals in excess salinity and excess temperature, rather than lagged correlations (lines 371-377). We have significantly expanded the discussion relating to this (lines 534-543) and included extra citations to relevant literature (Stott et al. 2008, Terray et al. 2012, Pierce et al. 2012, Skliris et al. 2014, line 541).

(b) Thank you for this suggestion, this is an excellent point. In the revised draft, we have explicitly calculated the redistribution of heat and salinity in the Atlantic through the Equator (lines 405-420), showing that our method is reliably capturing the redistribution of heat and salinity in response to AMOC changes. We have also added that AMOC is not the only relevant process at play (line 421-426).

(c) The original manuscript noted the resemblance of the excess salinity fields to the sea surface salinity changes in Zika et al. 2018 in response to an imposed heat flux, and also in response to water cycle amplification. We agree with the reviewer that there is some ambiguity in the text, so we have removed the reference to imposed water cycle amplification in the revised draft to improve this (line 534).

(d) We appreciate the reviewer's position and where appropriate we have maintained axis scales as constant; for example, in Figure 3 (Figure 7 in the revised draft), we have rescaled the plots in terms of mean temperature change in order to improve clarity. In some cases however, due to the different scales of each component and in each ocean basin, keeping axes scales constant would result in a number of important features becoming indiscernible. In these cases (Figures 6, 7, 8, 9) we have pointed out to the reader in the text and caption the different scales used. Specifically for the new Figure 7, we have also rewritten the surrounding text (lines 432-435) to make it clear that relationships are not emerging in every ocean basin, and that it is the North Atlantic that is of interest due to its different behaviour to every other ocean basin (lines 442-451).

In response to minor comments: The manuscript has been fundamentally rewritten to account for the deficiencies identified by the reviewers. We have attempted to identify and remove all redundancies, included additional description and information where this was lacking, and clarified the text where possible. We have added a significant number of references:

**Linking results to observations**

[revised manuscript text omitted]

Regarding Figure projections, we have reproduced all maps using a Robinson projection, rather than a native grid. Finally, following the reviewer's suggestion we have tried to improve the manuscript with respect to the terminology we use and its specificity, particularly regarding the type of variability we are considering (e.g. spatial/temporal etc) and the exact timescales being considered.

---

## Author Response (AR2)

**1 Reviewer 1 (Anonymous Referee 2 from revised submission)**

**Minor Comments**

- L46-48: Thank you for pointing this out - as the reviewer says, this should have read added/passive heat and anthropogenic storage locally. This has been rewritten to correct this.

- L68-69: This is a good point. The precise definitions of excess redistributed tracers were inadvertently removed from the rewritten theory section. This precise definition has been returned to the text, which should help to clarify the statement.

- L92-96: Thank you for pointing out this ambiguity - the text here has been rewritten to clarify this point. As the reviewer correctly identifies, our approach builds on the anticorrelation noted by Williams et al. 2021 between the background fields of temperature and DIC.

- L97-99: Again, thank you for pointing out this ambiguity - the text here has been rewritten to clarify the point being made.

- L100-103: We have reconfigured the text here to address potential ambiguities and better describe the methodological approach: specifically, we describe how the decomposition simply relies on the assumption that we can identify a clear spatial relationship between natural carbon and the variable we wish to decompose.

- L129: The reviewer is correct that $\Delta$ refers to changes from 1860 to an arbitrary time t. As noted by the reviewer, it is later defined, and we have moved the definition up in the text as suggested.

**Section 2.2**

- a) (Equation 12 (now 13)): Thank you for spotting this, this has been fixed.

- b) As the reviewer correctly points out, we are using the arguments of Williams et al. (2021): that the background temperature and carbon fields are generally strongly anticorrelated, and that we may assume that concentrations of natural carbon (Cnat) change only through redistribution. However, this anticorrelation varies spatially (through its soft tissue and carbonate components) and differences in the air-sea disequilibrium states of water masses when they are formed, so it is necessary to allow for this variation in order to improve estimates of local temperature (and salinity) redistribution. As suggested by the reviewer, we have added maps of $\kappa_r^T$ and $\kappa_r^S$ to Appendix D to illustrate their spatial variability, which we feel should help to address clarify the point raised. We have also rewritten the statement regarding the use of the word 'similarly': as the reviewer correctly points out, these parameterisations are only similar from the perspective of both encoding one of the arguments of Williams et al. 2021 each (the correlation of excess/anthropogenic carbon and excess temperature; and the anticorrelation of background/natural carbon and redistributed temperature): we have reworded this in response. Additionally, we have clarified that our use of subdecadal timescales in order to estimate $\kappa_r \hat{T}$ and $\kappa_r^S$ is in order to allow us to estimate the spatial covariability of the background fields, without it being necessary to have a decomposed temperature or salinity field at this point of the process. Regarding the appropriateness of this process over the RCP8.5 period, we have clarified that this should not be a problem due to the way in which the redistributed fields are designed and these correlations are calculated, with reference to the definitions mentioned in L68-69.

  As we have practical limitations on performing additional PAT experiments (see the point below on explicit validation), we are unable to show directly the correctness of Equation 7 (now Equation 8). However, we have instead added an appendix (E) in which we compare the estimates of total heat content change and heat content change due to redistribution in regions of the ocean which can be considered to be unventilated (determined using regions of the ocean where concentrations of anthropogenic carbon are negligible throughout the model run). Though this is of course an imperfect validation of Equation 7 (now 8), we are able to show that during the early 20th century, the high frequency (sub decadal) variability in the heat content of unventilated waters are very well captured

by the redistributed heat content of this region, and that over the 21st century, the ratio of the two asymptotes to approximately 0.8, indicating our method is reliably capturing at least 80% of the heat content change due to systematic redistribution as a result of climate change. We do note however that this is an imperfect validation due to the presence of other atmospheric forcings besides CO2, and so represents an upper limit of the uncertainty of our method: considering slightly shorter periods greatly improves the accuracy of the reconstruction, and we believe this is a bias introduced by the validation technique rather than the method.

Regarding the applicability of our technique to other tracers, the reviewer is correct to point out that we have not specifically shown that the method is applicable to other tracers. We have thus rewritten this statement to clarify that our point is merely that the assumptions and premises of our method are such that they can theoretically be applied to other tracers. The practicality of this will be investigated in further studies.

**2 Reviewer 2 (Anonymous referee 3):**

**Minor Comments**

- L14: Thank you for pointing this out, we have rephrased this and expanded context to make the statement clearer.

- L29: Thank you for pointing this out, this has been fixed. Full stops have been added to equations which require them, and we have replaced upper case with lower case theta as pointed out.

**Code availability & test cases**

As suggested by the reviewer, we have added a number of test cases and sample code/data to a GitHub repository (detailed and linked to at end of manuscript), following the standards of Irving et al. 2016. For ease of use and to minimise file sizes, we share a subset of the full dataset that enables a user to calculate a gamma value and decompose temperature for a single section of ocean (across the subtropical North Atlantic) for a single year, entirely reproducing our own decomposition for this region. We believe the available code should also enable users to apply our decomposition method to other simulations.

**Glossary / Table of Definitions**

The reviewer is correct to point out that a number of terms and concepts are introduced which may make the paper difficult to follow for a reader who is not already familiar with these terms  concepts. The reviewer suggests introducing a table of definitions / glossary in order to make the paper more accessible. As we have expanded upon the definition and interpretation of terms (as noted in our response to Reviewer 1), we feel that also including these definitions in a table within the main text would become repetitive. However, we agree with the reviewer that having definitions and descriptions grouped together would help improve the accessibility of the paper, and so we have added an Appendix containing the definitions and interpretation of these terms to the paper (Appendix F).

**Explicit Validation (PAT)**

We agree with both reviewers when they observe that it would improve the paper were we able to directly compute the errors on our estimates of temperature redistribution. Unfortunately, these simulations were performed >5 years ago on an HPC cluster which no longer exists. Rerunning the simulations and including a passive anomalous heat tracer is thus simply not practical due to the financial, infrastructure and time investment necessary. We hope that our theoretical justification and the favourable comparison with the validated alternative carbon based methodology of Bronselaer & Zanna 2020 supports the use of this new methodology to deconvolve the components of total heat and salinity changes.

---

## Author Response (AR3)

**Editors comments**

1. L231: Thank you for pointing this out, as suggested we have defined PCA here.

2. L274,377: This has been fixed, here and for the other occurrences in the text.

3. L302: Thank you, we have changed this as suggested.

4. L381: Thanks for noticing this, this has been fixed.

5. Figure 5: This has been fixed as per point 2.

6. L404: Thanks for spotting this, it has been fixed.

7. L456: Thanks for the suggestion, we have added it as suggested.

8. L459: This has been fixed as per point 6.

9. L484-485: Thank you for spotting this, this has been fixed here and at the other occurences through the text.

10. L796: There does not appear to be an update available to this paper.

11. L894: Thank you for pointing this out, we have updated this to the final citation.